# SHAPE-AWARE GRAPH SPECTRAL LEARNING

## ABSTRACT

Spectral Graph Neural Networks (GNNs) are gaining attention for their ability to surpass the limitations of message-passing GNNs. They rely on the supervision from the downstream task to learn spectral filters that capture the useful frequency information of the graph signal. However, some works empirically show that the preferred graph frequency is related to the graph homophily level. This relationship between graph frequency and graphs with homophily/heterophily has not been systematically analyzed and considered in existing spectral GNNs. To mitigate this gap, we conduct theoretical and empirical analyses which reveal that low-frequency importance is positively correlated with the homophily ratio, while high-frequency importance is negatively correlated. Motivated by this, we propose a shape-ware regularization on a Newton Interpolation-based spectral filter which can (i) learn arbitrary polynomial spectral filter and (ii) incorporate prior knowledge about the desired shape of the corresponding homophily level. Comprehensive experiments demonstrate that NewtonNet can achieve graph spectral filters with desired shapes and superior performance on both homophilous and heterophilous datasets. The code can be found at https://anonymous.4open.science/r/NewtonNet-8115.

## 1 INTRODUCTION

Graphs are pervasive in the real world, such as social networks (Wu et al., 2020), biological networks (Fout et al., 2017), and recommender systems (Wu et al., 2022). GNNs (Veličković et al.; Hamilton et al., 2017) have witnessed great success in various tasks such as node classification (Kipf & Welling, 2017), link prediction (Zhang & Chen, 2018), and graph classification (Zhang et al., 2018). Existing GNNs can be broadly categorized into spatial GNNs and spectral GNNs. Spatial GNNs (Veličković et al.; Klicpera et al., 2018; Abu-El-Haija et al., 2019) propagate and aggregate information from neighbors. Hence, it learns similar representations for connected neighbors. Spectral GNNs (Bo et al., 2023) learn a filter on the Laplacian matrix. The filter output is the amplitude of each frequency in the graph spectrum, which ultimately determines the learned representations.

In real-world scenarios, different graphs can have different homophily ratios. In homophilous graphs, nodes from the same class tend to be connected; while in heterophilous graphs, nodes from different classes tend to be connected. Several studies (Zhu et al., 2020a; Ma et al., 2022; Xu et al., 2022) have shown that heterophilous graphs pose a challenge for spatial GNNs based on message-passing as they implicitly assume the graph follows homophily assumption. Many works are proposed from the spatial perspective to address the heterophily problem (Zheng et al., 2022). Spectral GNNs (Bo et al., 2023; Levie et al., 2018) with learnable filters have shown great potential in addressing heterophily problems as they can learn spectral filters from the graph, which can alleviate the issue of aggregating noisy information from neighbors of different features/classes.

Recent works (Bo et al., 2021; Zhu et al., 2020a; Liu et al., 2022) have empirically shown that, in general, homophilous graphs contain more low-frequency information in graph spectrum and low-pass filter works well on homophilous graphs; while heterophilous graphs contain more high-frequency information and high-pass filter is preferred. However, they mostly provide empirically analysis without theoretical understanding. In addition, graphs can be in various homophily ratios instead of simply heterophily and homophily, which needs further analysis and understanding. However, *there lacks a systematic investigation and theoretical understanding between the homophily ratio of a graph and the frequency that would be beneficial for representation learning of the graph.*

To fill the gap, we systematically analyze the impact of amplitudes of each frequency on graphs with different homophily ratios. We observe that low-frequency importance is positively correlated with

the homophily ratio, while high-frequency importance is negatively correlated with the homophily ratio. Meanwhile, middle-frequency importance increases and then decreases as the homophily ratio increases. We also provide theoretical analysis to support our observations.

These observations suggest that an effective spectral GNN should be able to learn filters that can adapt to the homophily ratio of the graph, i.e., encouraging more important frequencies and discouraging frequencies with lower importance based on the graph ratio. However, this is a non-trivial task and existing spectral GNNs cannot satisfy the goal as they solely learn on downstream tasks and are not aware of the behaviors of the learned filter. As a result, existing spectral GNNs have the following problems: (i) they cannot adapt to varying homophily ratios, resulting in their inability to encourage and discourage different frequencies; and (ii) when only weak supervision is provided, there are not sufficient labels for the models to learn an effective filter, which degrades the performance.

To address the challenges, we propose a novel framework **NewtonNet**, which can learn filters that can encourage important frequencies while discourage non-important frequencies based on homophily ratio. Specifically, NewtonNet introduces a set of learnable points of filters supervised by label information, which gives the basic shape of the filter. It then adopts Newton Interpolation (Hildebrand, 1987) on those interpolation points to get the complete filter. As the points are learnable, NewtonNet can approximate arbitrary filters. As those interpolation points determine the shape of the filter, to adapt the filter to the homophily ratio, we design a novel **shape-aware regularization** on the points to encourage beneficial frequencies and discourage harmful frequencies to achieve an ideal filter shape. Experiments show that NewtonNet outperforms spatial and spectral GNNs on various datasets.

Our **main contributions** are: (1) We are the first to establish a well-defined relationship between graph frequencies and homophily ratios. We empirically and theoretically show that the more homophilous the graph is, the more beneficial the low-frequency is; while the more heterophilous the graph is, the more beneficial the high-frequency is. (2) We propose a novel framework NewtonNet using Newton Interpolation with shape-aware regularization that can learn better filter encourages beneficial frequency and discourages harmful frequency, resulting in better node representations. (3) Extensive experiments demonstrate the effectiveness of NewtonNet in various settings.

## 2 PRELIMINARIES

**Notations and Definitions.** Let $\mathcal{G} = (\mathcal{V}, \mathcal{E}, \mathbf{X})$ be an attributed undirected graph, where $\mathcal{V} = \{v_1, ..., v_N\}$ is the set of $N$ nodes, and $\mathcal{E} \subseteq \mathcal{V} \times \mathcal{V}$ is the set of edges. $\mathbf{X} = \{\mathbf{x}_1, ..., \mathbf{x}_N\} \in \mathbb{R}^{N \times F}$ is the node feature matrix, where $\mathbf{x}_i$ is the node features of node $v_i$ and $F$ is the feature dimension. $\mathbf{A} \in \mathbb{R}^{N \times N}$ is the adjacency matrix, where $\mathbf{A}_{ij} = 1$ if $(v_i, v_j) \in \mathcal{E}$; otherwise $\mathbf{A}_{ij} = 0$. $\mathcal{V}_{\mathcal{L}} = \mathcal{V} - \mathcal{V}_{\mathcal{U}} \subseteq \mathcal{V}$ is the training set with known class labels $\mathcal{Y}_L = \{y_v, \forall v \in \mathcal{V}_{\mathcal{L}}\}$, where $\mathcal{V}_{\mathcal{U}}$ is the unlabeled node sets. We use $f_\theta(\cdot)$ to denote the feature transformation function parameterized by $\theta$ and $g(\cdot)$ to denote the filter function. $\mathbf{D}$ is a diagonal matrix with $\mathbf{D}_{ii} = \sum_i \mathbf{A}_{ij}$. The normalized graph Laplacian matrix is given by $\mathbf{L} = \mathbf{I} - \mathbf{D}^{-1/2}\mathbf{A}\mathbf{D}^{-1/2}$. We use $(s, t)$ to denote a node pair and $\tilde{\mathbf{x}} \in \mathbb{R}^{N \times 1}$ to denote a graph signal, which can be considered as a feature vector.

Homophily Ratio measures the ratio of edges connecting nodes with the same label to all the edges, i.e., $h(G) = \frac{|\{(s,t) \in \mathcal{E} : y_s = y_t\}|}{|\mathcal{E}|}$. Graphs with high homophily and low heterophily have homophily ratios near 1, while those with low homophily and high heterophily have ratios near 0.

**Graph Spectral Learning.** For an undirected graph, the Laplacian matrix $\mathbf{L}$ is a positive semidefinite matrix. Its eigendecomposition is $\mathbf{L} = \mathbf{U}\mathbf{\Lambda}\mathbf{U}^\top$, where $\mathbf{U} = [\mathbf{u}_1, \cdots, \mathbf{u}_N]$ is the eigenvector matrix and $\mathbf{\Lambda} = \mathrm{diag}([\lambda_1, \cdots, \lambda_N])$ is the diagonal eigenvalue matrix. Given a graph signal $\tilde{\mathbf{x}}$, its graph Fourier transform is $\hat{\mathbf{x}} = \mathbf{U}^\top \tilde{\mathbf{x}}$, and inverse transform is $\tilde{\mathbf{x}} = \mathbf{U}\hat{\mathbf{x}}$. The graph filtering is given as

$$\mathbf{z} = \mathbf{U} \, \mathrm{diag}\left[g\left(\lambda_1\right), \ldots, g\left(\lambda_N\right)\right] \mathbf{U}^\top \tilde{\mathbf{x}} = \mathbf{U}g(\mathbf{\Lambda})\mathbf{U}^\top \tilde{\mathbf{x}}, \tag{1}$$

where $g$ is the spectral filter to be learned. However, eigendecomposition is computationally expensive with cubic time complexity. Thus, a polynomial function is usually adopted to approximate filters to avoid eigendecomposition, i.e., Eq. 1 is reformulated as a polynomial of $\mathbf{L}$ as

$$\mathbf{z} = \mathbf{U}g(\mathbf{\Lambda})\mathbf{U}^\top \tilde{\mathbf{x}} = \mathbf{U}\left(\sum\nolimits_{k=0}^{K} \theta_k \mathbf{\Lambda}^k\right)\mathbf{U}^\top \tilde{\mathbf{x}} = \left(\sum\nolimits_{k=0}^{K} \theta_k \mathbf{L}^k\right)\tilde{\mathbf{x}} = g(\mathbf{L})\tilde{\mathbf{x}}, \tag{2}$$

where $g$ is polynomial function and $\theta_k$ are the coefficients of the polynomial.

## 3 IMPACT OF FREQUENCIES ON GRAPHS WITH VARIOUS HOMOPHILY RATIOS

In this section, we first provide a theoretical analysis of the influence of different frequencies on graphs with various homophily ratios. We then perform preliminary experiments, which yield consistent results with our theoretical proof. Specifically, we find that low-frequency is beneficial while high-frequency is harmful to homophilous graphs; the opposite is true for heterophilous graphs. Our findings pave us the way for learning better representations based on the homophily ratio.

### 3.1 THEORETICAL ANALYSIS OF FILTER BEHAVIORS

Several studies (Bo et al., 2021; Zhu et al., 2020a) have empirically shown that low-frequency information plays a crucial role in homophilous graphs, while high-frequency information is more important for heterophilous graphs. However, none of them provide theoretical evidence to support this observation. To fill this gap, we theoretically analyze the relationship between homophily ratio and graph frequency. Specifically, we examine two graph filters that exhibit distinct behaviors in different frequency regions and explore their impacts on graphs with varying homophily ratios.

**Lemma 1.** *For a graph $\mathcal{G}$ with $N$ nodes, $C$ classes, and $N/C$ nodes for each class, if we randomly connect nodes to form the edge set $\mathcal{E}$, the expected homophily ratio is $\mathbb{E}(h(\mathcal{G})) = \frac{1}{C}$.*

The proof is in Appendix A.1. Lemma 1 reveals that if the edge set is constructed by random sampling, the expected homophily ratio of the graph is $1/C$. Hence, for a graph $\mathcal{G}$, if $h(\mathcal{G}) > 1/C$, it is more prone to generate homophilous edges than in the random case. If $h(\mathcal{G}) < 1/C$, heterophilous edges are more likely to form.

**Theorem 1.** *Let $\mathcal{G} = \{\mathcal{V}, \mathcal{E}\}$ be an undirected graph. $0 \leq \lambda_1 \cdots \leq \lambda_N$ are eigenvalues of its Laplacian matrix $\mathbf{L}$. Let $g_1$ and $g_2$ be two spectral filters satisfying the following two conditions: (1) $g_1(\lambda_i) < g_2(\lambda_i)$ for $1 \leq i \leq m$; and $g_1(\lambda_i) > g_2(\lambda_i)$ for $m + 1 \leq i \leq N$, where $1 < m < N$; and (2) They have the same norm of output values $\|[g_1(\lambda_1), \cdots, g_1(\lambda_N)]^\top\|_2^2 = \|[g_2(\lambda_1), \cdots, g_2(\lambda_N)]^\top\|_2^2$. For a graph signal $\mathbf{x}$, $\mathbf{x}^{(1)} = g_1(\mathbf{L})\mathbf{x}$ and $\mathbf{x}^{(2)} = g_2(\mathbf{L})\mathbf{x}$ are the corresponding representations after filters $g_1$ and $g_2$. Let $\Delta s = \sum_{(s,t)\in\mathcal{E}} \left[ (x_s^{(1)} - x_t^{(1)})^2 - (x_s^{(2)} - x_t^{(2)})^2 \right]$ be the difference between the total distance of connected nodes got by $g_1$ and $g_2$, where $x_s^1$ denotes the s-th element of $\mathbf{x}_s^{(1)}$. Then we have $\mathbb{E}[\Delta s] > 0$.*

Note that in the theorem, we assume $g_1$ and $g_2$ have the same norm to avoid trivial solutions. The proof of the theorem and the discussion of this assumption is in Appendix A.2. Theorem 1 reveals that if $g_1$ has higher amplitude in the high-frequency region and lower amplitude in the low-frequency region compared to $g_2$, it will result in less similarity in representations of connected nodes. In contrast, $g_2$ increases the representation similarity of connected nodes. As most edges in heterophilous graphs are heterophilous edges, $g_1$ is preferred to increase distances between heterophilously connected nodes. In contrast, homophilous graphs prefer $g_2$ to decrease distances between homophilously connected nodes. Next, we theoretically prove the claim that low-frequency is beneficial for the prediction of homophilous graphs, while high-frequency is beneficial for heterophilous graphs.

**Theorem 2.** *Let $\mathcal{G} = \{\mathcal{V}, \mathcal{E}\}$ be a balanced undirected graph with $N$ nodes, $C$ classes, and $N/C$ nodes for each class. $\mathcal{P}_{in}$ is the set of possible pairs of nodes from the same class. $\mathcal{P}_{out}$ is the set of possible pairs of nodes from different classes. $g_1$ and $g_2$ are two filters same as Theorem 1. Given an arbitrary graph signal $\mathbf{x}$, let $d_{in}^{(1)} = \sum_{(s,t)\in\mathcal{P}_{in}} \left( x_s^{(1)} - x_t^{(1)} \right)^2$ be the total intra-class distance, $d_{out}^{(1)} = \sum_{(s,t)\in\mathcal{P}_{out}} \left( x_s^{(1)} - x_t^{(1)} \right)^2$ be the total inter-class distance, and $\bar{d}_{in}^{(1)} = d_{in}^{(1)}/|\mathcal{P}_{in}|$ be the average intra-class distance while $\bar{d}_{out}^{(1)} = d_{out}^{(1)}/|\mathcal{P}_{out}|$ be the average inter-class distance. $\Delta \bar{d}^{(1)} = \bar{d}_{out}^{(1)} - \bar{d}_{in}^{(1)}$ is the difference between average inter-distance and intra-class distance. $d_{out}^{(2)}$, $d_{out}^{(2)}$, $\bar{d}_{in}^{(2)}$, $\bar{d}_{out}^{(2)}$, and $\Delta\bar{d}^{(2)}$ are corresponding values defined similarly on $\mathbf{x}^{(2)}$. If $\mathbb{E}[\Delta s] > 0$, we have: (1) when $h > \frac{1}{C}$, $\mathbb{E}[\Delta\bar{d}^{(1)}] < \mathbb{E}[\Delta\bar{d}^{(2)}]$; and (2) when $h < \frac{1}{C}$, $\mathbb{E}[\Delta\bar{d}^{(1)}] > \mathbb{E}[\Delta\bar{d}^{(2)}]$.*

The proof is in Appendix A.3. In Theorem 2, $\Delta\bar{d}$ shows the discriminative ability of the learned representation, where a large $\Delta\bar{d}$ indicates that representations of intra-class nodes are similar, while representations of inter-class nodes are dissimilar. As $g_1$ and $g_2$ are defined as described in Theorem 1, we can guarantee $\mathbb{E}[\Delta s] > 0$. Hence, homophilous graphs with $h > 1/C$ favor $g_1$; while

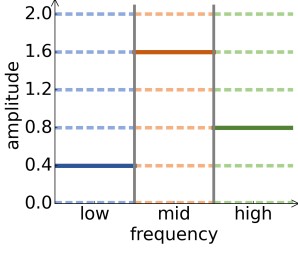
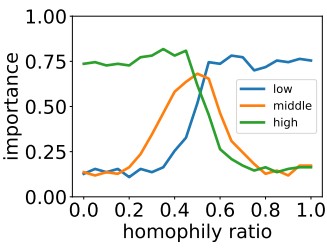

(a) Candidate filters

(b) Frequency importance

Figure 1: (a) Candidate filters. Blue, orange, and green dashed lines show the choices of amplitudes of low, middle, and high-frequency, respectively. We vary the amplitude of low, middle, and high-frequency among $\{0, 0.4, 0.8, 1.2, 1.6, 2.0\}$, which gives $6^3$ candidate filters. The solid line shows one candidate filter with $g(\lambda_{low} = 0.4)$, $g(\lambda_{mid} = 1.6)$ and $g(\lambda_{high}) = 0.8$. (b) The frequency importance of low, middle, and high on graphs with various homophily ratios.

heterophilous graphs with $h < 1/C$ favor $g_2$. This theorem shows that the *transition phase* of a balanced graph is $1/C$, where the transition phase is the point whether lower or higher frequencies are more beneficial changes.

Theorem 1 and Theorem 2 together guide us to the desired filter shape. When $h > 1/C$, the filter should involve more low-frequency and less high-frequency. When $h < 1/C$, the filter need to decrease the low-frequency and increase the high-frequency to be more discriminative.

### 3.2 EMPIRICAL ANALYSIS OF VARYING HOMOPHILY RATIOS

With the theoretical understanding, in this subsection, we further empirically analyze and verify the influence of high- and low- frequencies on node classification performance on graphs with various homophily ratios, which help us to design better GNN for node classification. As high-frequency and low-frequency are relative concepts, there is no clear division between them. Therefore, to make the granularity of our experiments better and to make our candidate filters more flexible, we divide the graph spectrum into three parts: low-frequency $0 \leq \lambda_{low} < \frac{2}{3}$, middle-frequency $\frac{2}{3} \leq \lambda_{mid} < \frac{4}{3}$, and high-frequency $\frac{4}{3} \leq \lambda_{high} \leq 2$. We perform experiments on synthetic datasets generated by contextual stochastic block model (CSBM) (Deshpande et al., 2018), which is a popular model that allows us to create synthetic attributed graphs with controllable homophily ratios (Ma et al., 2022; Chien et al., 2021). A detailed description of CSBM is in Appendix D.1.

To analyze which parts of frequencies in the graph spectrum are beneficial or harmful for graphs with different homophily ratios, we conduct the following experiments. We first generate synthetic graphs with different homophily ratios as $\{0, 0.05, 0.10, \cdots, 1.0\}$. Then for each graph, we conduct eigendecomposition as $\mathbf{L} = \mathbf{U \Lambda U}^\top$ and divide the eigenvalues into three parts: low-frequency $0 \leq \lambda_{low} < \frac{2}{3}$, middle-frequency $\frac{2}{3} \leq \lambda_{mid} < \frac{4}{3}$, and high-frequency $\frac{4}{3} \leq \lambda_{high} \leq 2$, because three-part provides more flexibility of the filter. As shown in Fig. 1(a), we vary the output values of the filter, i.e., amplitudes of $g(\lambda_{low})$, $g(\lambda_{mid})$, and $g(\lambda_{high})$ among $\{0, 0.4, 0.8, 1.2, 1.6, 2.0\}$ respectively, which leads to $6^3$ combinations of output values of the filter. Then we get transformed graph Laplacian $g(\mathbf{L}) = \mathbf{U}g(\mathbf{\Lambda})\mathbf{U}^\top$. For each filter, we use $g(\mathbf{L})$ as a convolutional matrix to learn node representation as $\mathbf{Z} = g(\mathbf{L})f_\theta(\mathbf{X})$, where $f_\theta$ is the feature transformation function implemented by an MLP. In summary, the final node representation is given by $\mathbf{z} = \mathbf{U} \operatorname{diag}\left[g(\lambda_{low}), g(\lambda_{mid}), g(\lambda_{high})\right]\mathbf{U}^\top f_\theta(\mathbf{x})$. By varying the amplitudes of each part of frequency, we vary how much a certain frequency is included in the representations. For example, the larger the value of $g(\lambda_{low})$ is, the more low-frequency information is included in $\mathbf{z}$. We then add a Softmax layer on top of $\mathbf{Z}$ to predict the label of each node. For each synthetic graph, we split the nodes into 2.5%/2.5%/95% for train/validation/test. For each filter, we conduct semi-supervised node classification on each synthetic graph and record the classification performance.

**Frequency Importance.** With the above experiments, to understand which filter works best for which homophily ratio, for each homophily ratio, we first select filters that give top 5% performance among the $6^3$ filters. Let $\mathcal{F}_h = \{[g_h^i(\lambda_{low}), g_h^i(\lambda_{mid}), g_h^i(\lambda_{high})]\}_{i=1}^K$ be the set of the best filters for homophily ratio $h$, where $[g_h^i(\lambda_{low}), g_h^i(\lambda_{mid}), g_h^i(\lambda_{high})]$ means the $i$-th best filter for $h$ and $K$

is the number of best filters. Then, importance scores of low, middle, and high frequency are given as

$$\mathcal{I}_h^{low} = \frac{1}{|\mathcal{F}_h|}\sum_{i=1}^{|\mathcal{F}_h|} g_h^i(\lambda_{low}), \quad \mathcal{I}_h^{mid} = \frac{1}{|\mathcal{F}_h|}\sum_{i=1}^{|\mathcal{F}_h|} g_h^i(\lambda_{mid}), \quad \mathcal{I}_h^{high} = \frac{1}{|\mathcal{F}_h|}\sum_{i=1}^{|\mathcal{F}_h|} g_h^i(\lambda_{high}). \quad (3)$$

The importance score of a certain frequency shows how much that frequency should be involved in the representation to achieve the best performance. The findings in Fig. 1(b) reveal two important observations: (i) as the homophily ratio increases, there is a notable increase in the importance of low-frequency, a peak and subsequent decline in the importance of middle-frequency, and a decrease in the importance of high-frequency; (ii) the transition phase occurs around the homophily ratio of $1/2$ as there are two distinct classes, resulting in a significant shift in the importance of frequencies. The frequency importance of graphs generated with other parameters is in Appendix E.3. These observations guide us in designing the spectral filter. *For high-homophily graphs, we want to preserve more low-frequency while removing high frequencies. For low-homophily graphs, more high frequencies and fewer low frequencies are desired. For graphs with homophily ratios near the transition phase, we aim the model to learn adaptively.*

## 4 THE PROPOSED FRAMEWORK NEWTONNET

The preceding theoretical and emperical analysis show that the decision to include more or less of a specific frequency is contingent upon the homophily ratio. Hence, it is desirable for the model to learn this ratio and encourage or discourage low, middle, or high-frequency accordingly. However, existing spectral methods either use predefined filters or let the model freely

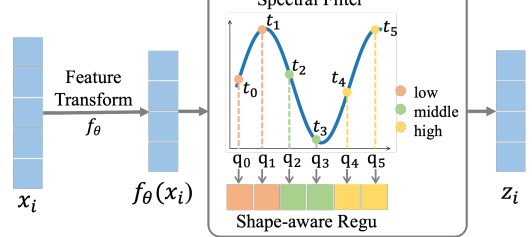

Figure 2: The overall framework.

learn a filter solely based on labels, which lacks an efficient mechanism to promote or discourage low, middle, or high-frequency based on homophily ratio, especially when the label is sparse. Therefore, the regularization of spectral filter values poses a significant challenge for existing methods. To address the challenge, we propose a novel framework NewtonNet, which can regularize spectral filter values and encourage or discourage different frequencies according to the homophily ratio. The basic idea of NewtonNet is to introduce some points $\{(q_0, t_0), \cdots, (q_n, t_K)\}$ of the graph filter, where $\{q_i\}$ are fixed and $\{t_i\}$ are learnable. Fig. 2 shows the case when $K = 5$. These points give the basic shape of the filter, and we adopt Newton Interpolation to approximate the filter function based on these points. As those points are learnable, we can approximate any filters. To incorporate the prior knowledge, we propose a novel shape-aware regularization to regularize those learnable points to adapt to the homophily ratio of the graph. The filter is then used to learn node representation and optimized for node classification. Next, we introduce the details.

### 4.1 NEWTON INTERPOLATION AND NEWTONNET

In order to learn flexible filters and incorporate prior knowledge in Section 3 in the filter, i.e., encouraging or discouraging different frequencies based on the homophily ratio, we first propose to adopt a set of $K + 1$ learnable points $\mathcal{S} = \{(q_0, t_0), \cdots, (q_n, t_K)\}$ of the filter. Interpolated points $\mathbf{q} = [q_0, \cdots, q_K]$ are fixed and distributed equally among low, middle, and high-frequency depending on their values. The corresponding values $\mathbf{t} = [t_0, \cdots, t_K]$ are learnable and updated in each epoch of training. These learnable points gives the basic shape of a filter. Then we can use interpolation method to approximate a filter function $g$ by passing these points such that $g(q_k) = t_k$, where $0 \leq k \leq K$. This gives us two advantages: (**i**) As $\mathbf{t}$ determines the basic shape of a filter $g$ and is learnable, we can learn arbitrary filter by learning $\mathbf{t}$ that minimizes the node classification loss; and (**ii**) As the filter $g$ is determined by $\mathbf{t}$, it allows us to add regularization on $\mathbf{t}$ to encourage beneficial and discourage harmful frequencies based on homophily ratio.

Specifically, with these points, we adopt Newton Interpolation (Hildebrand, 1987) to learn the filter, which is a widely-used method for approximating a function by fitting a polynomial on a set of given points. Given $K + 1$ points and their values $\mathcal{S} = \{(q_0, t_0), \cdots, (q_n, t_K)\}$ of an unknown function $g$, where $q_k$ are pairwise distinct, $g$ can be interpolated based on Newton Interpolation as follows.

**Definition 1** (Divided Differences). *The divided differences $\hat{g}_{\mathbf{t}}$ defined on $\mathbf{t}$ are given recursively as*

$$\hat{g}_{\mathbf{t}}[q_k] = t_k, \qquad\qquad\qquad\qquad\qquad 0 \le k \le K$$

$$\hat{g}_{\mathbf{t}}[q_k, \cdots, q_{k+j}] = \frac{\hat{g}_{\mathbf{t}}[q_{k+1}, \cdots, q_{k+j}] - \hat{g}_{\mathbf{t}}[q_k, \cdots, q_{k+j-1}]}{q_{k+j} - q_k}, \quad 1 \le j \le K, 0 \le k \le K - j. \tag{4}$$

**Lemma 2** (Newton Interpolation). *$g(x)$ can be interpolated by Newton interpolation as:*

$$g(q) \approx \hat{g}(q) = \sum_{k=0}^{K} [a_k n_k(q)] = \sum_{k=0}^{K} \left\{ \hat{g}_{\mathbf{t}}[q_0, \cdots, q_k] \prod_{i=0}^{k-1} (q - q_i) \right\}, \tag{5}$$

*where $n_k(q) = \prod_{i=0}^{k-1} (q - q_i)$ are Newton polynomial basis and $a_k = \hat{g}_{\mathbf{t}}[q_0, \cdots, q_k]$ are coefficients.*

Eq. 5 satisfies $g(q_k) = t_k$ and $K$ is the power of the polynomial. As mentioned in Eq. 2, we directly apply the spectral filter $g$ on Laplacian $\mathbf{L}$ to get the final representations. Then Eq. 5 is reformulated as $g(\mathbf{L}) = \sum_{k=0}^{K} \left\{ \hat{g}_{\mathbf{t}}[q_0, \cdots, q_k] \prod_{i=0}^{k-1} (\mathbf{L} - q_i \mathbf{I}) \right\}$, where $\mathbf{I}$ is the identity matrix.

Following (Chien et al., 2021), to increase the expressive power, the node features are firstly transformed by a transformation layer $f_\theta$, then a convolutional matrix learned by NewtonNet is applied to transformed features, which is shown in Fig. 2. Mathematically, NewtonNet can be formulated as:

$$\mathbf{Z} = \sum_{k=0}^{K} \left\{ \hat{g}_{\mathbf{t}}[q_0, \cdots, q_k] \prod_{i=0}^{k-1} (\mathbf{L} - q_i \mathbf{I}) \right\} f_\theta(\mathbf{X}), \tag{6}$$

where $f_\theta$ is a feature transformation function, and we use a two-layer MLP in this paper. $q_0, \cdots, q_K$ can be any points $\in [0, 2]$. We use the equal-spaced points, i.e., $q_i = 2i/K$. $q_i$ controls different frequencies depending on its value. $t_0, \cdots, t_K$ are learnable filter output values and are randomly initialized. By learning $t_i$, we are not only able to learn arbitrary filters, but can also apply regularization on $t_i$'s to adapt the filter based on homophily ratio, which will be discussed in Section 4.2. With the node representation $\mathbf{z}_v$ of node $v$, $v$'s label distribution probability is predicted as $\hat{\mathbf{y}}_v = \text{softmax}(\mathbf{z}_v)$.

## 4.2 SHAPE-AWARE REGULARIZATION

As $\{t_0, \cdots, t_k\}$ determines the shape of filter $g$, we can control the filter shape by adding constraints on learned function values. As shown in Fig. 2, we slice $\mathbf{t}$ into three parts

$$\mathbf{t}_{low} = [t_0, \cdots, t_{i-1}], \ \mathbf{t}_{mid} = [t_i, \cdots, t_{j-1}], \ \mathbf{t}_{high} = [t_j, \cdots, t_K], \ 0 < i < j < K, \tag{7}$$

which represent the amplitude of low, middle, and high-frequency, respectively. In this paper, we set $K = 5, i = 2, j = 4$ so that we have six learnable points and two for each frequency. According to the analysis results in Section 3, as the homophily ratio increases, we design the following regularizations. (**i**) For low-frequency, the amplitude should decrease, and we discourage low-frequency before the transition phase and encourage it after the transition phase. Thus, we add a loss term as $\left( \frac{1}{C} - h \right) \|\mathbf{t}_{low}\|_2^2$ to regularize the amplitudes of low-frequency. When the homophily ratio is smaller than the transition phase $1/C$, the low-frequency is harmful; therefore, the loss term has a positive coefficient. In contrast, when the homophily ratio is larger than the transition phase, the low-frequency is beneficial, and we give a positive coefficient for the loss term. (**ii**) For high-frequency, the amplitude should decrease, and we encourage the high-frequency before the transition phase and discourage it after the transition phase. Similarly, the regularization for high-frequency is $\left( h - \frac{1}{C} \right) \|\mathbf{t}_{high}\|_2^2$. (**iii**) For middle-frequency, we observe that it has the same trend as the low-frequency before the transition phase and has the same trend as the high-frequency after the transition phase in Fig. 1 (b). Therefore, we have the regularization $\left| h - \frac{1}{C} \right| \|\mathbf{t}_{mid}\|_2^2$. These three regularization work together to determine the shape based on homophily ratio $h$. Then the shape-aware regularization can be expressed as

$$\min_{\theta, \mathbf{t}} \mathcal{L}_{SR} = \gamma_1 \left( \frac{1}{C} - h \right) \|\mathbf{t}_{low}\|_2^2 + \gamma_2 \left| h - \frac{1}{C} \right| \|\mathbf{t}_{mid}\|_2^2 + \gamma_3 \left( h - \frac{1}{C} \right) \|\mathbf{t}_{high}\|_2^2, \tag{8}$$

where $\gamma_1, \gamma_2, \gamma_3 > 0$ are scalars to control contribution of the regularizers, respectively. $h$ is the learned homophily ratio, calculated and updated according to the labels learned in every epoch. The final objective function of NewtonNet is

$$\min_{\theta, \mathbf{t}} \mathcal{L} = \mathcal{L}_{CE} + \mathcal{L}_{SR}, \tag{9}$$

where $\mathcal{L}_{CE} = \sum_{v \in \mathcal{V}^L} \ell(\hat{\boldsymbol{y}}_v, \boldsymbol{y}_v)$ is classification loss on labeled nodes and $\ell(\cdot, \cdot)$ denotes cross entropy loss. The training algorithm of NewtonNet is in Appendix B.

**Compared with other approximation and interpolation methods** This Newton Interpolation method has its advantages compared with approximation methods, e.g., ChebNet (Defferrard et al., 2016), GPRGNN (Chien et al., 2021), and BernNet (He et al., 2021b). Both interpolation and approximation are common methods to approximate a polynomial function. However, the interpolation function passes all the given points accurately, while approximation methods minimize the error between the function and given points. We further discuss the difference between interpolation and approximation methods in Appendix C. This difference makes us be able to readily apply shape-aware regularization on learned $\mathbf{t}$. However, approximation methods do not pass given points accurately, so we cannot make points learnable to approximate arbitrary functions and apply regularization.

Compared to other interpolation methods like ChebNetII (He et al., 2022), Newton interpolation offers greater flexibility in choosing interpolated points. In Eq. 6, $q_i$ values can be freely selected from the range [0, 2], allowing adaptation to specific graph properties. In contrast, ChebNetII uses fixed Chebyshev points for input values, limiting precision in narrow regions and requiring increased complexity (larger K) to address this limitation. Additionally, ChebNetII can be seen as a special case of NewtonNet when Chebyshev points are used as $q_i$ values (He et al., 2022).

**Complexity Analysis** NewtonNet exhibits a time complexity of $O(KEF + NF^2)$ and a space complexity of $O(KE + F^2 + NF)$, where $E$ represents the number of edges. Notably, the complexity scales linearly with $K$. A detailed analysis is provided in Appendix B.1.

## 5 RELATED WORK

**Spatial GNNs.** Existing GNNs can be categorized into spatial and spectral-GNNs. Spatial GNNs (Kipf & Welling, 2016; Veličković et al.; Klicpera et al., 2018; Hamilton et al., 2017) adopt message-passing mechanism, which updates a node's representation by aggregating the message from its neighbors. For example, GCN (Kipf & Welling, 2017) uses a weighted average of neighbors' representations as the aggregate function. APPNP (Klicpera et al., 2018) first transforms features and then propagates information via personalized PageRank. However, The message-passing GNNs (Gilmer et al., 2017) rely on the homophily and thus fail on heterophilous graphs as they smooth over nodes with different labels and make the representations less discriminative. Many works (Zhu et al., 2020a; Xu et al., 2022; Abu-El-Haija et al., 2019; He et al., 2021a; Wang et al., 2021; Dai et al., 2022; Zhu et al., 2021a) design network structures from a spatial perspective to address this issue. H$_2$GCN (Zhu et al., 2020a) aggregates information for ego and neighbor representations separately instead of mixing them together. LW-GCN (Dai et al., 2022) proposes the label-wise aggregation strategy to preserve information from heterophilous neighbors. Some other works (Veličković et al., 2019; Zhu et al., 2020b; Xu et al., 2021; Xiao et al.) also focus on self-supervised learning on graphs.

**Spectral GNNs.** Spectral GNNs (Bo et al., 2023; 2021; Li et al., 2021; Zhu et al., 2021b) learn a polynomial function of graph Laplacian served as the convolutional matrix. Recently, more works consider the relationship between heterophily and graph frequency. To name a few, Bo et al. (2021) maintains a claim that besides low-frequency signal, the high-frequency signal is also useful for heterophilous graphs. Li et al. (2021) states that high-frequency components contain important information for heterophilous graphs. Zhu et al. (2020a) claims high frequencies contain more information about label distributions than low frequencies. Therefore, learnable spectral GNNs (He et al., 2021b; 2022; Chien et al., 2021; Levie et al., 2018) that can approximate arbitrary filters perform inherently well on heterophily. The methods used to approximate the polynomial function vary among them, such as the Chebyshev polynomial for ChebNet (Defferrard et al., 2016), Bernstein polynomial for BernNet (He et al., 2021b), Jacobi polynomial for JacobiConv (Wang & Zhang, 2022), and Chebyshev interpolation for ChebNetII (He et al., 2022). Other spectral methods design non-polynomial filters or transform the basis (Bo et al., 2023).

## 6 EXPERIMENTS

In this section, we conduct experiments to evaluate the effectiveness of the proposed NewtonNet and address the following research questions: **RQ1** How effective is NewtonNet on datasets of

Table 1: Node classification performance (Accuracy(%) ± Std.) under full-supervised setting

| | Cora | Cite. | Pubm. | Cham. | Squi. | Croc. | Texas | Corn. | Penn94 | Gamer | Genius |
|---|---|---|---|---|---|---|---|---|---|---|---|
| MLP | 73.28±1.9 | 70.95±2.1 | 86.08±0.7 | 48.86±2.3 | 32.27±1.0 | 65.37±1.0 | 75.79±8.4 | 75.79±8.4 | 74.18±0.3 | 65.24±0.2 | 86.80±0.1 |
| GCN | 87.86±2.1 | 75.47±1.0 | 87.00±0.6 | 66.12±3.7 | 54.65±2.7 | 72.95±0.6 | 54.21±10.2 | 54.21±6.5 | 83.23±0.2 | 66.58±0.2 | 80.28±0.0 |
| Mixhop | 87.81±1.7 | 74.32±1.3 | 88.50±0.7 | 64.39±0.6 | 49.95±1.9 | 73.63±0.8 | 70.00±8.7 | 73.16±2.6 | 84.09±0.2 | 67.27±0.2 | 88.39±0.4 |
| APPNP | 89.04±1.5 | 77.04±1.4 | 88.84±0.4 | 56.60±1.7 | 37.00±1.5 | 67.42±0.9 | 76.84±4.7 | 80.53±4.7 | 75.91±0.2 | 66.76±0.2 | 87.19±0.2 |
| ChebNet | 88.32±2.0 | 75.47±1.0 | 89.62±0.3 | 62.94±2.2 | 43.07±0.7 | 72.01±1.0 | 81.05±3.9 | 82.63±5.7 | 82.63±0.3 | 67.57±0.2 | 86.69±0.2 |
| GPRGNN | 89.20±1.6 | 77.48±1.9 | 89.50±0.4 | 71.15±2.1 | 55.18±1.3 | 69.68±1.0 | 86.37±1.1 | 83.16±4.9 | 84.08±0.2 | 64.44±0.3 | 87.41±0.1 |
| BernNet | 89.76±1.6 | 77.49±1.4 | 89.47±0.4 | 72.19±1.6 | 55.43±1.1 | 69.70±0.9 | 85.26±6.4 | 84.21±8.6 | 83.04±0.1 | 62.90±0.2 | 86.52±0.1 |
| ChebNetII | 88.51±1.5 | 75.83±1.3 | 89.51±0.6 | 69.91±2.3 | 52.83±0.8 | 67.86±1.6 | 84.74±3.1 | 81.58±8.0 | 83.52±0.2 | 62.53±0.2 | 86.49±0.1 |
| JacobiConv | 88.98±0.7 | 75.76±1.9 | 89.55±0.5 | 73.87±1.6 | 57.56±1.8 | 67.69±1.1 | 84.17±6.8 | 75.56±6.1 | 83.28±0.1 | 67.68±0.2 | 88.03±0.4 |
| GloGNN++ | 88.11±1.8 | 74.68±1.3 | 89.12±0.2 | 73.94±1.8 | 56.58±1.7 | 69.25±1.1 | 82.22±4.5 | 81.11±4.4 | 84.94±0.2 | 67.50±0.3 | 89.31±0.1 |
| NewtonNet | 89.39±1.4 | 77.87±1.9 | 89.68±0.5 | 74.47±1.5 | 61.58±0.8 | 75.70±0.4 | 87.11±3.8 | 86.58±5.3 | 84.56±0.1 | 67.92±0.3 | 88.20±0.1 |

various domains and sizes with varying degrees of homophily and heterophily? **RQ2** How does NewtonNet compare to other baselines under weak supervision? **RQ3** To what extent does shape-aware regularization contribute to the performance of NewtonNet? **RQ4** Is the learned filter shape consistent with our analysis on homophilous and heterophilous graphs?

## 6.1 EXPERIMENTAL SETUP

We use node classification to evaluate the performance. Here we briefly introduce the dataset, baselines, and settings in the experiments. We give details in Appendix D.

**Datasets.** To evaluate NewtonNet on graphs with various homophily ratios, we adopt three homophilous datasets (Sen et al., 2008): Cora, Citeseer, Pubmed, and six heterophilous datasets (Pei et al., 2020; Rozemberczki et al., 2021; Traud et al., 2012; Lim et al., 2021), Chameleon, Squirrel, Crocodile, Texas, Cornell, Penn94, Twitch-gamer, and Genius. Details are in Appendix D.2.

**Baselines.** We compare NewtonNet with representative baselines, including (i) non-topology method: MLP; (ii) spatial methods: GCN (Kipf & Welling, 2017), Mixhop (Abu-El-Haija et al., 2019), APPNP (Klicpera et al., 2018), GloGNN++ (Li et al., 2022); and (iii) spectral methods: ChebNet (Defferrard et al., 2016), GPRGNN (Chien et al., 2021), BernNet (He et al., 2021b), ChebNetII (He et al., 2022), JacobiConv (Wang & Zhang, 2022). Details of methods are in Appendix D.3.

**Settings.** We adopt the commonly-used split of 60%/20%/20% for train/validation/test sets. For a fair comparison, for each method, we select the best configuration of hyperparameters using the validation set and report the mean accuracy and variance of 10 random splits on the test.

## 6.2 NODE CLASSIFICATION PERFORMANCE ON HETEROPHILY AND HOMOPHILY GRAPHS

Table 1 shows the results on node classification, where boldface denotes the best results and underline denotes the second-best results. We observe that learnable spectral GNNs outperform other baselines since spectral methods can learn beneficial frequencies for prediction under supervision, while spatial methods and predetermined spectral methods only obtain information from certain frequencies. NewtonNet achieves state-of-the-art performance on eight of nine datasets with both homophily and heterophily, and it achieves the second-best result on Cora. This is because NewtonNet efficiently learns the filter shape through Newton interpolation and shape-aware regularization.

## 6.3 PERFORMANCE WITH WEAK SUPERVISION

As we mentioned before, learning filters solely based on the downstream tasks may lead to sub-optimal performance, especially when there are rare labels. Thus, this subsection delves into the impact of the training ratio on the performance of various GNNs. We keep the validation and test set ratios fixed at 20%, and vary the training set ratio from 0.1 to 0.6. We select several best-performing baselines and plot their performance on the Chameleon and Squirrel datasets in Fig. 3.

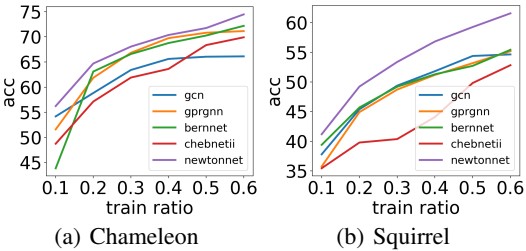

(a) Chameleon    (b) Squirrel

Figure 3: The accuracy on Chameleon and Squirrel datasets as the training set ratio varies.

From the figure, we observe: (1) When the training ratio is large, spectral methods perform well because they have sufficient labels to learn a filter shape. Conversely, when the training ratio is small, some spectral methods do not perform as well as GCN as there are not enough labels for the models to learn the filter; (2) NewtonNet consistently outperforms all baselines. When the training ratio is

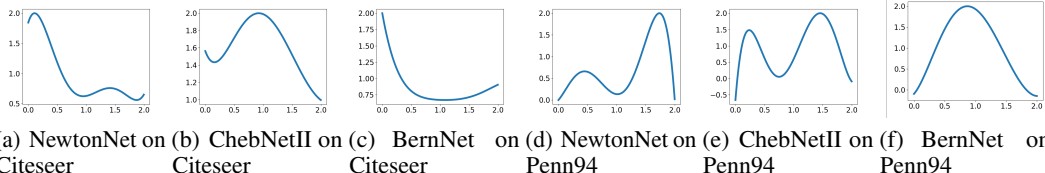

(a) NewtonNet on Citeseer  (b) ChebNetII on Citeseer  (c) BernNet on Citeseer  (d) NewtonNet on Penn94  (e) ChebNetII on Penn94  (f) BernNet on Penn94

Figure 5: Learned filters on Citeseer and Penn94.

large, NewtonNet can filter out harmful frequencies while encouraging beneficial ones, leading to representations of higher quality. When the training ratio is small, the shape-aware regularization of NewtonNet incorporates prior knowledge to provide guidance in learning better filters.

### 6.4 ABLATION STUDY

Next, we study the effect of shape-aware regularization on node classification performance. We show the best result of NewtonNet with and without shape-aware regularization, where we use the same search space as in Section 6.2. Fig. 4 gives the results on five datasets. From the figure, we can observe that the regularization contributes more to the performance on heterophilous datasets (Chameleon, Squirrel, Crocodile) than on homophilous datasets (Cora, Citeseer). The reason is as follows. Theorem 2 and Fig. 1(b) show that the importance of frequencies changes significantly near the transition phase, $1/C$. Cora and Citeseer

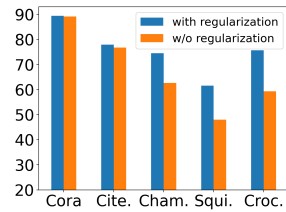

Figure 4: Ablation Study

have homophily ratios of 0.81 and 0.74, while they have 7 and 6 classes, respectively. Since Cora and Citeseer are highly homophilous datasets whose homophily ratios are far from the transition faces, almost only low frequencies are beneficial, and thus the filter shapes are easier to learn. By contrast, the homophily ratios of Chameleon, Squirrel, and Crocodile are 0.24, 0.22, and 0.25, respectively, while they have five classes. Their homophily ratios are around the transition phase; therefore, the model relies more on shape-aware regularization to guide the learning process. We also conduct the hyperparameter analysis for $\gamma_1$, $\gamma_2$, $\gamma_3$, and $K$ in Appendix E.1.

### 6.5 ANALYSIS OF LEARNED FILTERS

Here we present the learned filters of NewtonNet on Citeseer and Chameleon and compare them with those learned by BernNet and ChebNetII. In Fig. 5, we plot the average filters by calculating the average temperatures of 10 splits. On the homophilous graph Citeseer, NewtonNet encourages low-frequency and filters out middle and high-frequency, which is beneficial for the representations of homophilous graphs. In contrast, ChebNetII and BernNet do not incorporate any shape-aware regularization, and their learning processes include middle and high-frequency, harming the representations of homophilous graphs. On the heterophilous dataset Penn94, the filter learned by NewtonNet contains more high-frequency and less-frequency, while harmful frequencies are included in the filters learned by ChebNetII and BernNet. These results reveal that NewtonNet, with its shape-aware regularization, can learn a filter shape with beneficial frequencies while filtering out harmful frequencies. More results of the learned filters can be found in Appendix E.4. We also present the learned homophily ratio in Appendix E.2.

### 7 CONCLUSION AND FUTURE WORK

In this paper, we propose a novel framework NewtonNet for learning spectral filters GNNs. NewtonNet incorporates prior knowledge about the desired shape of spectral filters based on the homophily ratio of the dataset. The empirical and theoretical analysis reveals that low-frequency is positively correlated with the homophily ratio, while high-frequency is negatively correlated. NewtonNet utilizes Newton Interpolation with shape-aware regularization to learn arbitrary polynomial spectral filters that adapt to different homophily levels. Experimental results on real-world datasets demonstrate the effectiveness of the proposed framework. In the paper, we propose one possible regularization for filter and we leave the exploration of other regularizers as future work.

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

# A   DETAILED PROOFS

## A.1   PROOF OF LEMMA 1

**Lemma 1.** *For a graph $\mathcal{G}$ with $N$ nodes, $C$ classes, and $N/C$ nodes for each class, if we randomly connect nodes to form the edge set $\mathcal{E}$, the expected homophily ratio is $\mathbb{E}(h(\mathcal{G})) = \frac{1}{C}$.*

*Proof.* We first randomly sample a node $s$ from $\mathcal{V}$, assuming its label is $y_s$. Then we sample another node $t$. Because the classes are balanced, we have

$$
\begin{aligned}
\mathrm{P}(y_s = y_t) &= \frac{1}{C} , \\
\mathrm{P}(y_s \neq y_t) &= \frac{C-1}{C} .
\end{aligned}
\tag{10}
$$

Therefore, if each node pair in $\mathcal{E}$ is sampled randomly, we have

$$
\mathbb{E}(h) = \frac{1}{|\mathcal{E}|} \cdot |\mathcal{E}| \cdot \frac{1}{C} = \frac{1}{C},
\tag{11}
$$

which completes the proof. $\qquad\square$

## A.2   PROOF OF THEOREM 1

**Theorem 1.** *Let $\mathcal{G} = \{\mathcal{V}, \mathcal{E}\}$ be an undirected graph. $0 \leq \lambda_1 \cdots \leq \lambda_N$ are eigenvalues of its Laplacian matrix $\mathbf{L}$. Let $g_1$ and $g_2$ be two spectral filters satisfying the following two conditions: (1) $g_1(\lambda_i) < g_2(\lambda_i)$ for $1 \leq i \leq m$; and $g_1(\lambda_i) > g_2(\lambda_i)$ for $m+1 \leq i \leq N$, where $1 < m < N$; and (2) They have the same norm of output values $\|[g_1(\lambda_1), \cdots, g_1(\lambda_N)]^\top\|_2^2 = \|[g_2(\lambda_1), \cdots, g_2(\lambda_N)]^\top\|_2^2$. For a graph signal $\mathbf{x}$, $\mathbf{x}^{(1)} = g_1(\mathbf{L})\mathbf{x}$ and $\mathbf{x}^{(2)} = g_2(\mathbf{L})\mathbf{x}$ are the corresponding representations after filters $g_1$ and $g_2$. Let $\Delta s = \sum_{(s,t)\in\mathcal{E}} \left[ (x_s^{(1)} - x_t^{(1)})^2 - (x_s^{(2)} - x_t^{(2)})^2 \right]$ be the difference between the total distance of connected nodes got by $g_1$ and $g_2$, where $x_s^1$ denotes the $s$-th element of $\mathbf{x}_s^{(1)}$. Then we have $\mathbb{E}[\Delta s] > 0$.*

*Proof.* **1.** Given the eigendecomposition of Laplacian $\mathbf{L} = \mathbf{U}\boldsymbol{\Lambda}\mathbf{U}^\top$, because $\mathbf{u}_0, \cdots, \mathbf{u}_{N-1}$ are orthonormal eigenvectors, any unit graph signal $\mathbf{x}$ can be expresses as the linear combination of the eigenvectors:

$$
\mathbf{x} = \sum_{i=1}^{N} c_i \mathbf{u}_i,
\tag{12}
$$

where $c_i = \mathbf{u}_i^T \mathbf{x}$ are the coefficients of each eigenvector. Then, we have

$$
\begin{aligned}
\mathbf{x}^{(1)} &= g_1(\mathbf{L})\mathbf{x} = \mathbf{U}g_1(\boldsymbol{\Lambda})\mathbf{U}^\top \mathbf{x} = \left( \sum_{i=1}^{N} g_1(\lambda_i)\mathbf{u}_i\mathbf{u}_i^\top \right) \left( \sum_{i=1}^{N} c_i\mathbf{u}_i \right) = \sum_{i=1}^{N} g_1(\lambda_i)c_i\mathbf{u}_i, \\
\mathbf{x}^{(2)} &= g_2(\mathbf{L})\mathbf{x} = \mathbf{U}g_2(\boldsymbol{\Lambda})\mathbf{U}^\top \mathbf{x} = \left( \sum_{i=1}^{N} g_2(\lambda_i)\mathbf{u}_i\mathbf{u}_i^\top \right) \left( \sum_{i=1}^{N} c_i\mathbf{u}_i \right) = \sum_{i=1}^{N} g_2(\lambda_i)c_i\mathbf{u}_i.
\end{aligned}
\tag{13}
$$

Note that $\lambda_i$ and $u_i$ are the eigenvalues and eigenvectors of original Laplacian $\mathbf{L}$. Moreover, we have

$$
\mathbf{c} = \mathbf{U}^\top \mathbf{x} \qquad c_i = \mathbf{u}_i^\top \mathbf{x}.
\tag{14}
$$

Eq. 14 demonstrates that each element of $\mathbf{c}$ is determined independently by the product of each eigenvalue and $\mathbf{x}$. We have $-1 = -\|\mathbf{u}_i\|_2\|\mathbf{x}\|_2 \leq c_i^2 \leq \|\mathbf{u}_i\|_2\|\mathbf{x}\|_2 = 1$. Furthermore, because $x$ is an arbitrary unit graph signal, it can achieve any value with $\|\mathbf{x}\|_2 = 1$. It's reasonable for us to assume that $c_i$'s are independently identically distributed with mean 0.

**2.** For any graph signal $\mathbf{x}$, its smoothness is the total distance between the connected nodes, which is given by,

$$\sum_{(s,t)\in\mathcal{E}} (x_s - x_t)^2 = \mathbf{x}^\top \mathbf{L}\mathbf{x},$$

$$\sum_{(s,t)\in\mathcal{E}} (x_s^{(1)} - x_t^{(2)})^2 = \mathbf{x}^{(1)\top}\mathbf{L}\mathbf{x}^{(1)}, \tag{15}$$

$$\sum_{(s,t)\in\mathcal{E}} (x_s^{(2)} - x_t^{(2)})^2 = \mathbf{x}^{(2)\top}\mathbf{L}\mathbf{x}^{(2)}.$$

Note that the smoothness score of an eigenvector equals the corresponding eigenvalue:

$$\lambda_i = \mathbf{u}_i^\top \mathbf{L}\mathbf{u}_i = \sum_{(s,t)\in\mathcal{E}} (u_{i,s} - u_{i,t})^2. \tag{16}$$

Then we plug in Eq. 12 and Eq. 13 into Eq. 15 to get,

$$\sum_{(s,t)\in\mathcal{E}} (x_s - x_t)^2 = \left(\sum_{i=1}^{N} c_i \mathbf{u}_i^\top\right)\left(\sum_{i=1}^{N} \lambda_i \mathbf{u}_i \mathbf{u}_i^\top\right)\left(\sum_{i=1}^{N} c_i \mathbf{u}_i\right) = \sum_{i=1}^{N} c_i^2 \lambda_i, \tag{17}$$

$$\sum_{(s,t)\in\mathcal{E}} (x_s^{(1)} - x_t^{(1)})^2 = \left(\sum_{i=1}^{N} g_1(\lambda_i)c_i \mathbf{u}_i^\top\right)\left(\sum_{i=1}^{N} \lambda_i \mathbf{u}_i \mathbf{u}_i^\top\right)\left(\sum_{i=1}^{N} g_1(\lambda_i)c_i \mathbf{u}_i\right) = \sum_{i=1}^{N} c_i^2 \lambda_i g_1^2(\lambda_i).$$

$$\tag{18}$$

$$\sum_{(s,t)\in\mathcal{E}} (x_s^{(2)} - x_t^{(2)})^2 = \left(\sum_{i=1}^{N} g_2(\lambda_i)c_i \mathbf{u}_i^\top\right)\left(\sum_{i=1}^{N} \lambda_i \mathbf{u}_i \mathbf{u}_i^\top\right)\left(\sum_{i=1}^{N} g_2(\lambda_i)c_i \mathbf{u}_i\right) = \sum_{i=1}^{N} c_i^2 \lambda_i g_2^2(\lambda_i).$$

$$\tag{19}$$

**3.** For i.i.d. random variables $c_i$ and any $i < j$, we have

$$\mathbb{E}[c_i^2] = \mathbb{E}[c_j^2]$$
$$\Rightarrow \lambda_i \mathbb{E}[c_i^2] = \mathbb{E}[\lambda_i c_i^2] \leq \mathbb{E}[\lambda_j c_j^2] = \lambda_j \mathbb{E}[c_j^2] \tag{20}$$

We are interested in the expected difference between the total distance of connected nodes got by $g_1$ and $g_2$. Let $\Delta s$ denote difference between the total distance of connected nodes got by $g_1$ and $g_2$, i.e.,

$$\Delta s = \sum_{(s,t)\in\mathcal{E}} \left[(x_s^{(1)} - x_t^{(1)})^2 - (x_s^{(2)} - x_t^{(2)})^2\right] \tag{21}$$

Then, the expected difference between the total distance of connected nodes got by $g_1$ and $g_2$ is

$$\mathbb{E}\left[\Delta s\right] = \mathbb{E}\left[\sum_{(s,t)\in\mathcal{E}} (x_s^{(1)} - x_t^{(1)})^2\right] - \mathbb{E}\left[\sum_{(s,t)\in\mathcal{E}} (x_s^{(2)} - x_t^{(2)})^2\right]$$

$$= \mathbb{E}\left[\sum_{i=1}^{N} c_i^2 \lambda_i g_1^2(\lambda_i)\right] - \mathbb{E}\left[\sum_{i=1}^{N} c_i^2 \lambda_i g_2^2(\lambda_i)\right] \tag{22}$$

$$= \sum_{i=1}^{N} \left\{\left[g_1^2(\lambda_i) - g_2^2(\lambda_i)\right] \lambda_i \mathbb{E}\left[c_i^2\right]\right\}$$

**4.** We assume $\|[g_1(\lambda_1), \cdots, g_1(\lambda_N)]^\top\|_2^2 = \|[g_2(\lambda_1), \cdots, g_2(\lambda_N)]^\top\|_2^2$ so that $g_1$ and $g_2$ have the same $\ell_2$-norms. We make this assumption to avoid some trivial solutions. For example, if we simply multiply the representation $\mathbf{x}$ with a constant, the value in Eq. 17 will also be enlarged and reduced,

but the discriminative ability is unchanged. Therefore, we have

$$\sum_{i=1}^{N} g_1^2(\lambda_i) = \sum_{i=1}^{N} g_2^2(\lambda_i)$$

$$\Rightarrow \sum_{i=1}^{m} g_1^2(\lambda_i) + \sum_{i=m+1}^{N} g_1^2(\lambda_i) = \sum_{i=1}^{m} g_2^2(\lambda_i) + \sum_{i=m+1}^{N} g_2^2(\lambda_i) \tag{23}$$

$$\Rightarrow 0 < \sum_{i=1}^{m} \left[g_2^2(\lambda_i) - g_1^2(\lambda_i)\right] = \sum_{i=m+1}^{N} \left[g_1^2(\lambda_i) - g_2^2(\lambda_i)\right],$$

because $g_1(\lambda_i) < g_2(\lambda_i)$ for $1 \le i \le m$ and $g_1(\lambda_i) > g_2(\lambda_i)$ for $m+1 \le i \le N$. By applying the results in Eq. 23 and Eq. 20, we have

$$0 < \sum_{i=1}^{m} \left\{ \left[g_2^2(\lambda_i) - g_1^2(\lambda_i)\right] \lambda_i \right\} < \lambda_m \sum_{i=1}^{m} \left[g_2^2(\lambda_i) - g_1^2(\lambda_i)\right]$$

$$< \lambda_{m+1} \sum_{i=1}^{m} \left[g_2^2(\lambda_i) - g_1^2(\lambda_i)\right] = \lambda_{m+1} \sum_{i=m+1}^{N} \left[g_1^2(\lambda_i) - g_2^2(\lambda_i)\right] \tag{24}$$

$$< \sum_{i=m+1}^{N} \left\{ \left[g_1^2(\lambda_i) - g_2^2(\lambda_i)\right] \lambda_i \right\}.$$

Then we can derive that

$$\sum_{i=1}^{N} \left\{ \left[g_1^2(\lambda_i) - g_2^2(\lambda_i)\right] \lambda_i \right\} > 0$$

$$\Rightarrow \sum_{i=1}^{N} \left\{ \left[g_1^2(\lambda_i) - g_2^2(\lambda_i)\right] \lambda_i \mathbb{E}\left[c_i^2\right] \right\} = \mathbb{E}\left[\Delta s\right] > 0 \tag{25}$$

$$\Rightarrow \mathbb{E}\Big[ \sum_{(s,t)\in\mathcal{E}} (x_s^{(1)} - x_t^{(1)})^2 \Big] > \mathbb{E}\Big[ \sum_{(s,t)\in\mathcal{E}} (x_s^{(2)} - x_t^{(2)})^2 \Big]$$

which completes our proof. □

### A.3 PROOF OF THEOREM 2

**Theorem 2.** *Let $\mathcal{G} = \{\mathcal{V}, \mathcal{E}\}$ be a balanced undirected graph with $N$ nodes, $C$ classes, and $N/C$ nodes for each class. $\mathcal{P}_{in}$ is the set of possible pairs of nodes from the same class. $\mathcal{P}_{out}$ is the set of possible pairs of nodes from different classes. $g_1$ and $g_2$ are two filters same as Theorem 1. Given an arbitrary graph signal $\mathbf{x}$, let $d_{in}^{(1)} = \sum_{(s,t)\in\mathcal{P}_{in}} \left(x_s^{(1)} - x_t^{(1)}\right)^2$ be the total intra-class distance, $d_{out}^{(1)} = \sum_{(s,t)\in\mathcal{P}_{out}} \left(x_s^{(1)} - x_t^{(1)}\right)^2$ be the total inter-class distance, and $\bar{d}_{in}^{(1)} = d_{in}^{(1)}/|\mathcal{P}_{in}|$ be the average intra-class distance while $\bar{d}_{out}^{(1)} = d_{out}^{(1)}/|\mathcal{P}_{out}|$ be the average inter-class distance. $\Delta\bar{d}^{(1)} = \bar{d}_{out}^{(1)} - \bar{d}_{in}^{(1)}$ is the difference between average inter-distance and intra-class distance. $d_{out}^{(2)}$, $d_{out}^{(2)}$, $\bar{d}_{in}^{(2)}$, $\bar{d}_{out}^{(2)}$, and $\Delta\bar{d}^{(2)}$ are corresponding values defined similarly on $\mathbf{x}^{(2)}$. If $\mathbb{E}[\Delta s] > 0$, we have: (1) when $h > \frac{1}{C}$, $\mathbb{E}[\Delta\bar{d}^{(1)}] < \mathbb{E}[\Delta\bar{d}^{(2)}]$; and (2) when $h < \frac{1}{C}$, $\mathbb{E}[\Delta\bar{d}^{(1)}] > \mathbb{E}[\Delta\bar{d}^{(2)}]$.*

*Proof.* In graph $\mathcal{G}$, the number of possible homophilous (intra-class) edges (including self-loop) is,

$$|\mathcal{P}_{in}| = \frac{C}{2} \frac{N}{C} \left(\frac{N}{C}\right) = \frac{N^2}{2C}. \tag{26}$$

The number of possible heterophilous (inter-class) edges is,

$$|\mathcal{P}_{out}| = \frac{C}{2} \frac{N}{C} \left(\frac{C-1}{C}N\right) = \frac{N^2}{2}\left(\frac{C-1}{C}\right). \tag{27}$$

Therefore, we have

$$\bar{d}_{in}^{(i)} = \frac{d_{in}^{(i)}}{|\mathcal{P}_{in}|} = \frac{2Cd_{in}^{(i)}}{N^2} \,, \quad i \in \{1,2\} \tag{28}$$

$$\bar{d}_{out}^{(i)} = \frac{d_{out}^{(i)}}{|\mathcal{P}_{out}|} = \frac{2Cd_{out}^{(i)}}{N^2(C-1)} \,, \quad i \in \{1,2\} \tag{29}$$

$$
\begin{aligned}
\Delta\bar{d}^{(i)} &= \bar{d}_{out}^{(i)} - \bar{d}_{in}^{(i)} \\
&= \frac{2Cd_{out}^{(i)}}{N^2(C-1)} - \frac{2Cd_{in}^{(i)}}{N^2} \\
&= \frac{2C}{(C-1)N^2} \left[ d_{out}^{(i)} - (C-1)d_{in}^{(i)} \right]
\end{aligned}
\tag{30}
$$

$\mathcal{E}_{in}$ denotes the set of edges connecting nodes from the same class (intra-class edges). $\mathcal{E}_{out}$ denotes the set of edges connecting nodes from different classes (inter-class edges). There are $|\mathcal{E}|$ edges, $h \cdot |\mathcal{E}|$ homophilous edges, and $(1-h) \cdot |\mathcal{E}|$ heterophilous edges. In expectation, each edge has the same difference of the distance of connected nodes got by $g_1$ and $g_2$, i.e.,

$$
\begin{aligned}
\mathbb{E}\left[\Delta s\right] &= \mathbb{E}\left[ \sum_{(s,t)\in\mathcal{E}} \left[ (x_s^{(1)} - x_t^{(1)})^2 - (x_s^{(2)} - x_t^{(2)})^2 \right] \right] \\
\mathbb{E}\left[h\Delta s\right] &= \mathbb{E}\left[ \sum_{\substack{(s,t)\in\mathcal{E} \\ y_s = y_t}} \left[ (x_s^{(1)} - x_t^{(1)})^2 - (x_s^{(2)} - x_t^{(2)})^2 \right] \right] \\
\mathbb{E}\left[(1-h)\Delta s\right] &= \mathbb{E}\left[ \sum_{\substack{(s,t)\in\mathcal{E} \\ y_s \neq y_t}} \left[ (x_s^{(1)} - x_t^{(1)})^2 - (x_s^{(2)} - x_t^{(2)})^2 \right] \right]
\end{aligned}
\tag{31}
$$

If we solely consider the direct influence of graph convolution on connected nodes, the relationship between $d'$ and $d$ can be expressed as follows:

$$\mathbb{E}\left[ \bar{d}_{out}^{(1)} - \bar{d}_{out}^{(2)} \right] = (1-h)\mathbb{E}\left[\Delta s\right] \quad \text{and} \quad \mathbb{E}\left[ \bar{d}_{in}^{(1)} - \bar{d}_{in}^{(2)} \right] = h\mathbb{E}\left[\Delta s\right] \tag{32}$$

$$
\begin{aligned}
\mathbb{E}\left[ \Delta\bar{d}^{(1)} - \Delta\bar{d}^{(2)} \right] &= \mathbb{E}\left[ \frac{2C}{(C-1)N^2} \left[ (d_{out}^{(1)} - d_{out}^{(2)}) - (C-1)(d_{in}^{(1)} - d_{in}^{(2)}) \right] \right] \\
&= \frac{2C}{(C-1)N^2} \left[ (1-h)\mathbb{E}[\Delta s] - (C-1)h\mathbb{E}[\Delta s] \right] \\
&= \frac{2C}{(C-1)N^2} \left[ \mathbb{E}[\Delta s](1 - Ch) \right]
\end{aligned}
\tag{33}
$$

Because $\frac{2C}{(C-1)N^2} > 0$, then we have
(1) when $h > \frac{1}{C}$, if $\Delta s < 0$, then $\mathbb{E}[\Delta\bar{d}^{(1)}] > \mathbb{E}[\Delta\bar{d}^{(2)}]$; if $\Delta s > 0$, then $\mathbb{E}[\Delta\bar{d}^{(1)}] < \mathbb{E}[\Delta\bar{d}^{(2)}]$.
(2) when $h < \frac{1}{C}$, if $\Delta s < 0$, then $\mathbb{E}[\Delta\bar{d}^{(1)}] < \mathbb{E}[\Delta\bar{d}^{(2)}]$; if $\Delta s > 0$, then $\mathbb{E}\Delta\bar{d}^{(1)}] > \mathbb{E}[\Delta\bar{d}^{(2)}]$. $\quad\square$

## B  TRAINING ALGORITHM OF NEWTONNET

We show the training algorithm of NewtonNet in Algorithm 1. We first randomly initialize $\theta$, $h$, and $\{t_0, \cdots, t_K\}$. We update $h$ according to the predicted labels of each iteration. We update the learned representations and $\theta$, and $\{t_0, \cdots, t_K\}$ accordingly until convergence or reaching max iteration.

---

**Algorithm 1** Training Algorithm of NewtonNet

---

**Input:** $\mathcal{G} = (\mathcal{V}, \mathcal{E}, \mathbf{X}), \mathcal{Y}_L, K, \{q_0, \cdots, q_K\}, \gamma_1, \gamma_2, \gamma_3,$
**Output:** $\theta, h, \{t_0, \cdots, t_K\}$
 1: Randomly initialize $\theta, h, \{t_0, \cdots, t_K\}$
 2: **repeat**
 3:    Update representations $\mathbf{z}$ by Eq. 6
 4:    Update predicted labels $\hat{\boldsymbol{y}}_v$
 5:    Update $h$ with predicted labels
 6:    $\mathcal{L} \leftarrow$ Eq. 9
 7:    Update $\theta$ and $\{t_0, \ldots, t_K\}$
 8: **until** *convergence or reaching max iteration*

---

## B.1 COMPLEXITY ANALYSIS

**Time complexity.** According to Blakely et al. (2021), the time complexity of GCN is $O(L(EF + NF^2))$, where $E$ is the number of edges, and $L$ is the number of layers. One-layer GCN has the formula $\mathbf{X}^{l+1} = \sigma(\mathbf{A}\mathbf{X}^l\mathbf{W}^l)$. $O(NF^2)$ is the time complexity of feature transformation, and $O(EF)$ is the time complexity of neighborhood aggregation. Following GPRGNN, ChebNetII, and BernNet, NewtonNet uses the "first transforms features and then propagates" strategy. According to Eq. 6, the feature transformation part is implemented by an MLP with time $O(NF^2)$. And the propagation part has the time complexity with $O(KEF)$. In other words, NewtonNet has a time complexity $O(KEF + NF^2)$, which is linear to $K$. BernNet's time complexity is quadratic to $K$. We summarize the time complexity in Table 2.

Table 2: Time and space complexity.

| Method | Time Complexity | Space Complexity |
|---|---|---|
| MLP | $O(NF^2)$ | $O(NF^2)$ |
| GCN | $O(L(EF + NF^2))$ | $O(E + LF^2 + LNF)$ |
| GPRGNN | $O(KEF + NF^2)$ | $O(E + F^2 + NF)$ |
| ChebNetII | $O(KEF + NF^2)$ | $O(E + F^2 + NF)$ |
| BernNet | $O(K^2EF + NF^2)$ | $O(KE + F^2 + NF)$ |
| NewtonNet | $O(KEF + NF^2)$ | $O(KE + F^2 + NF)$ |

Table 3: Running Time (ms/epoch) of each method.

| Method | Chameleon | Pubmed | Penn94 | Genius |
|---|---|---|---|---|
| MLP | 1.909 | 2.283 | 6.119 | 10.474 |
| GCN | 2.891 | 3.169 | 22.043 | 20.159 |
| Mixhop | 3.609 | 4.299 | 19.702 | 27.041 |
| GPRGNN | 4.807 | 4.984 | 10.572 | 12.522 |
| ChebNetII (K=5) | 4.414 | 4.871 | 9.609 | 12.189 |
| ChebNetII (K=10) | 7.352 | 7.447 | 13.661 | 15.346 |
| BernNet (K=5) | 8.029 | 11.730 | 19.719 | 20.168 |
| BernNet (K=10) | 20.490 | 20.869 | 49.592 | 43.524 |
| NewtonNet (K=5) | 6.6135 | 7.075 | 17.387 | 17.571 |
| NewtonNet (K=10) | 12.362 | 13.042 | 25.194 | 30.836 |

To examine our analysis, Table 3 shows the running time of each method. We employ 5 different masks with 2000 epochs and calculate the average time of each epoch. We observe that (1) For the spectral methods, NewtonNet, ChebNetII, and GPRGNN run more quickly than BernNet since their time complexity is linear to $K$ while BernNet is quadratic; (2) NewtonNet cost more time than GPRGNN and ChebNetII because it calculates a more complex polynomial; (3) On smaller datasets (Chameleon, Pubmed), GCN runs faster than NewtonNet. On larger datasets (Penn94, Genius), NewtonNet and other spectral methods are much more efficient than GCN. This is because we only

transform and propagate the features once, but in GCN, we stack several layers to propagate to more neighbors. In conclusion, NewtonNet is scalable on large datasets.

**Space complexity.** Similarly, GCN has a space complexity $O(E + LF^2 + LNF)$. $O(F^2)$ is for the weight matrix of feature transformation while $O(NF)$ is for the feature matrix. $O(E)$ is caused by the sparse edge matrix. NewtonNet has the space complexity $O(KE + F^2 + NF)$ because it pre-calculates $(\mathbf{L} - q_i\mathbf{I})$ in Equation 6. We compare the space complexity in Table 2 and the memory used by each model in Table 4. On smaller datasets, NewtonNet has a similar space consumption with GCN. However, NewtonNet and other spectral methods are more space efficient than GCN on larger datasets because we do not need to stack layers. Therefore, NewtonNet has excellent scalability.

Table 4: Memory usage (MB) of each method.

| Method | Chameleon | Pubmed | Penn94 | Genius |
|---|---|---|---|---|
| MLP | 1024 | 1058 | 1862 | 1390 |
| GCN | 1060 | 1114 | 3320 | 2012 |
| Mixhop | 1052 | 1124 | 2102 | 2536 |
| GPRGNN | 1046 | 1060 | 1984 | 1370 |
| ChebNetII | 1046 | 1080 | 1982 | 1474 |
| BernNet | 1046 | 1082 | 2026 | 1544 |
| NewtonNet | 1048 | 1084 | 2292 | 1868 |

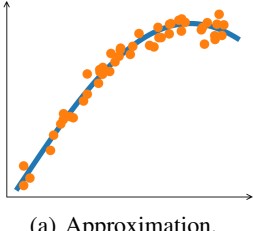
(a) Approximation.

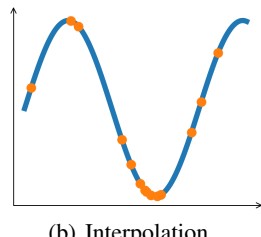
(b) Interpolation.

Figure 6: Difference between approximation and interpolation.

## C  APPROXIMATION VS INTERPOLATION

Approximation and interpolation are two common curve fitting methods (Hildebrand, 1987). Given an unknown continuous function $\hat{f}(x)$, and its values at $n + 1$ known points $\{(x_0, \hat{f}(x_0)), \cdots, (x_n, \hat{f}(x_n))\}$, we want to use $g(x)$ to fit unknown $\hat{f}(x)$. Approximation methods aim to minimize the error between the original function and estimated function $|\hat{f}(x) - g(x)|$; while interpolation methods aim to fit the data and make $\hat{f}(x_i) = g(x_i), i = 0, \cdots, n$. In other words, the interpolation function passes every known point exactly, but the approximation function finds minimal error among the known points. Fig. 6 shows the difference. The property of interpolation allows us to learn the function values directly in our model, which is discussed in Section 4.

## D  EXPERIMENTAL DETAILS

### D.1  SYNTHETIC DATASETS

In this paper, we employ contextual stochastic block model (CSBM) (Deshpande et al., 2018) to generate synthetic datasets. CSBM provides a model to generate synthetic graphs with controllable inter-class and intra-class edge probability. It assumes features of nodes from the same class conform to the same distribution. Assume there are two equal-size classes $\{c_0, c_1\}$ with $n$ nodes for each class. CSBM generates edges and features by:

$$\mathbb{P}(\mathbf{A}_{ij} = 1) = \begin{cases} \frac{1}{n}(d + \sigma\sqrt{d}) & \text{when } y_i = y_j \\ \frac{1}{n}(d - \sigma\sqrt{d}) & \text{when } y_i \neq y_j \end{cases} \qquad \mathbf{x}_i = \sqrt{\frac{\mu}{n}}v_i\mathbf{u} + \frac{\mathbf{w}_i}{\sqrt{F}} \qquad (34)$$

where $d$ is the average node degree, $\mu$ is mean value of Gaussian distribution, $F$ is the feature dimension, entries of $\mathbf{w}_i \in \mathbb{R}^p$ has independent standard normal distributions, and $\mathbf{u} \sim \mathcal{N}(0, I_F/F)$. We can control homophily ratio $h$ by changing $\sigma = \sqrt{d}(2h-1)$, $-\sqrt{d} \leq \sigma \leq \sqrt{d}$. When $\sigma = -\sqrt{d}$, it is a totally heterophilous graph; when $\sigma = \sqrt{d}$, it is a totally homophilous graph. Following (Chien et al., 2021), we adopt $d = 5, \mu = 1$ in this paper. We vary $\sigma$ to generate graphs with different homophily levels. In Fig. 1(b), we adopt $2n = 3000, F = 3000$ to generate the synthetic dataset. We vary the number of nodes $2n$ and number of features $F$ to generate different CSBM datasets and show their frequency importance in Fig. 9.

## D.2 REAL-WORLD DATASETS

**Citation Networks** (Sen et al., 2008): Cora, Citeseer, and PubMed are citation network datasets. Cora consists of seven classes of machine learning papers, while CiteSeer has six. Papers are represented by nodes, while citations between two papers are represented by edges. Each node has features defined by the words that appear in the paper's abstract. Similarly, PubMed is a collection of abstracts from three types of medical papers.

**WebKB** (Pei et al., 2020): Cornell, Texas, and Wisconsin are three sub-datasets of WebKB. They are collected from a set of websites of several universities' CS departments and processed by (Pei et al., 2020). For each dataset, a node represents a web page and an edge represents a hyperlink. Node features are the bag-of-words representation of each web page. We aim to classify the nodes into one of the five classes, student, project, course, staff, and faculty.

**Wikipedia Networks** (Rozemberczki et al., 2021): Chameleon, Squirrel, and Crocodile are three topics of Wikipedia page-to-page networks. Articles from Wikipedia are represented by nodes, and links between them are represented by edges. Node features indicate the occurrences of specific nouns in the articles. Based on the average monthly traffic of the web page, the nodes are divided into five classes.

**Social Networks** (Lim et al., 2021): Penn94 (Traud et al., 2012) is a social network of friends among university students on Facebook in 2005. The network consists of nodes representing individual students, each with their reported gender identified. Additional characteristics of the nodes include their major, secondary major/minor, dormitory or house, year of study, and high school attended.

Twitch-gamers (Rozemberczki & Sarkar, 2021) is a network graph of Twitch accounts and their mutual followers. Node attributes include views, creation date, language, and account status. The classification is binary and to predict whether the channel has explicit content.

Genius (Lim & Benson, 2021) is from the genius.com social network, where nodes represent users and edges connect mutual followers. Node attributes include expertise scores, contribution counts, and user roles. Some users are labeled "gone," often indicating spam. Our task is to predict these marked nodes.

Table 5: Statistics of real-world datasets.

| Dataset | Citation | | | Wikipedia | | | WebKB | | Social | | |
|---|---|---|---|---|---|---|---|---|---|---|---|
| | Cora | Cite. | Pubm. | Cham. | Squi. | Croc. | Texas | Corn. | Penn94 | Genius | Gamer |
| Nodes | 2708 | 3327 | 19717 | 2277 | 5201 | 11,631 | 183 | 183 | 41554 | 421,961 | 168,114 |
| Edges | 5429 | 4732 | 44338 | 36101 | 217,073 | 360040 | 309 | 295 | 1,362,229 | 984,979 | 6,797,557 |
| Attributes | 1433 | 3703 | 500 | 2325 | 2089 | 128 | 1703 | 1703 | 4814 | 12 | 7 |
| Classes | 7 | 6 | 3 | 5 | 5 | 5 | 5 | 5 | 2 | 2 | 2 |
| $h$ | 0.81 | 0.74 | 0.80 | 0.24 | 0.22 | 0.25 | 0.11 | 0.31 | 0.47 | 0.62 | 0.55 |

## D.3 BASELINES

We compare our method with various state-of-the-art methods for both spatial and spectral methods. First, we compare the following spatial methods:

- **MLP**: Multilayer Perceptron predicts node labels using node attributes only without incorporating graph structure information.
- **GCN** (Kipf & Welling, 2017): Graph Convolutional Network is one of the most popular MPNNs using 1-hop neighbors in each layer.

- **MixHop** (Abu-El-Haija et al., 2019): MixHop mixes 1-hop and 2-hop neighbors to learn higher-order information.
- **APPNP** (Klicpera et al., 2018): APPNP uses the Personalized PageRank algorithm to propagate the prediction results of GNN to increase the propagation range.
- **GloGNN++** (Li et al., 2022): GloGNN++ is a method for creating node embeddings by aggregating information from global nodes in a graph using coefficient matrices derived through optimization problems.

We also compare with recent state-of-the-art spectral methods:

- **ChebNet** (Defferrard et al., 2016): ChebNet uses Chebyshev polynomial to approximate the filter function. It is a more generalized form of GCN.
- **GPRGNN** (Chien et al., 2021): GPRGNN uses Generalized PageRank to learn weights for combining intermediate results.
- **BernNet** (He et al., 2021b): ChebNet uses Bernstein polynomial to approximate the filter function. It can learn arbitrary target functions.
- **ChebNetII** (He et al., 2022): ChebNet uses Chebyshev interpolation to approximate the filter function. It mitigates the Runge phenomenon and ensures the learned filter has a better shape.
- **JacobiConv** (Wang & Zhang, 2022): JacobiConv uses Jacobi basis to study the expressive power of spectral GNNs.

### D.4 SETTINGS

We run all of the experiments with 10 random splits and report the average performance with the standard deviation. For full-supervised learning, we use 60%/20%/20% splits for the train/validation/test set. For a fair comparison, for each method, we select the best configuration of hyperparameters using the validation set and report the mean accuracy and variance of 10 random splits on the test. For NewtonNet, we choose $K = 5$ and use a MLP with two layers and 64 hidden units for encoder $f_\theta$. We search the learning rate of encoder and propagation among {0.05, 0.01, 0.005}, the weight decay rate among {0, 0.0005}, the dropout rate for encoder and propagation among {0.0, 0.1, 0.3, 0.5, 0.7, 0.9}, and $\gamma_1, \gamma_2, \gamma_3$ among {0, 1, 3, 5}. For other baselines, we use the original code and optimal hyperparameters from authors if available. Otherwise, we search the hyperparameters within the same search space of NewtonNet.

## E ADDITIONAL EXPERIMENTAL RESULTS

### E.1 HYPERPARAMETER ANALYSIS

In our hyperparameter sensitivity analysis on the Citeseer and Chameleon datasets, we investigated the effects of varying the values of $\gamma_1$, $\gamma_2$, and $\gamma_3$ among {0, 0.01, 0.1, 1, 10, 100}. The accuracy results were plotted in Figure 7. We made the following observations. For the Chameleon dataset, increasing the value of $\gamma_1$ resulted in improved performance, as it effectively discouraged low-frequency components. As for $\gamma_2$, an initial increase led to performance improvements since it balanced lower and higher frequencies. However, further increases in $\gamma_2$ eventually led to a decline in performance. On the other hand, increasing $\gamma_3$ had a positive effect on performance, as it encouraged the inclusion of more high-frequency components.

Regarding the Citeseer dataset, we found that increasing the values of $\gamma_1$, $\gamma_2$, and $\gamma_3$ initially improved performance. However, there was a point where further increases in these regularization terms caused a decrease in performance. This can be attributed to the fact that excessively large regularization terms overshadowed the impact of the cross entropy loss, thus hindering the model's ability to learn effectively. We also observe that the change of Chameleon is more than that in Citeseer, so heterophilous graphs need more regularization.

We also investigate the sensitivity of the parameter $K$. While keeping the remaining optimal hyperparameters fixed, we explore different values of $K$ within the set {2, 5, 8, 11, 14}. The corresponding accuracy results are presented in Fig. 8. In both datasets, we observe a pattern of

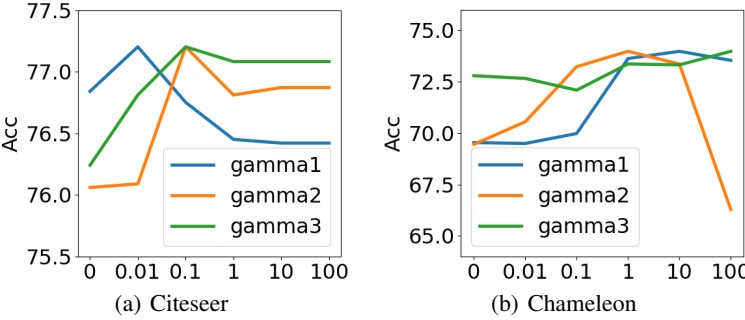

Figure 7: Hyperparameter Analysis for $\gamma_1$, $\gamma_2$, and $\gamma_3$

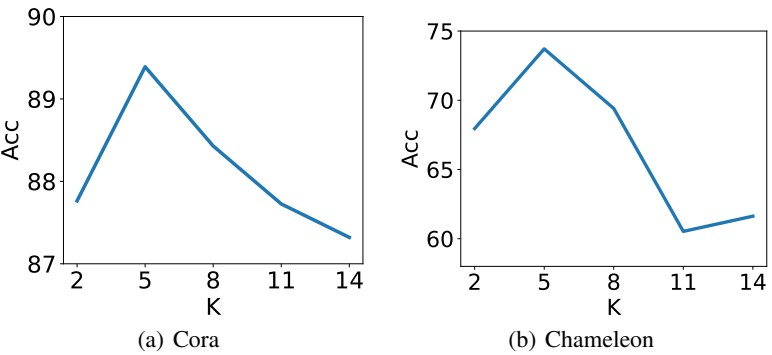

Figure 8: Hyperparameter Analysis for $K$.

increasing performance followed by a decline. This behavior can be attributed to the choice of $K$. If $K$ is set to a small value, the polynomial lacks the necessary power to accurately approximate arbitrary functions. Conversely, if $K$ is set to a large value, the polynomial becomes susceptible to the Runge phenomenon (He et al., 2022).

### E.2 LEARNED HOMOPHILY RATIO

Table 6 presents the real homophily ratio alongside the learned homophily ratio for each dataset. The close proximity between the learned and real homophily ratios indicates that our model can estimate the homophily ratio accurately so that it can further guide the learning of spectral filter.

### E.3 MORE RESULTS OF FREQUENCY IMPORTANCE

In Fig. 9, we present more results of frequency importance on CSBM datasets with different numbers of nodes and features. We fix $d = 5$ and $\mu = 1$ in Eq. 34 and vary the number of nodes and features among $\{800, 1000, 1500, 2000, 3000\}$. We can get similar conclusions as in Section 3.2.

### E.4 LEARNED FILTERS

The learned filters of NewtonNet, BernNet, and ChebNetII for each dataset are illustrated in Fig. 10 to Fig. 18. Our observations reveal that NewtonNet exhibits a distinct ability to promote or discourage specific frequencies based on the homophily ratio. In the case of homophilous datasets, NewtonNet emphasizes low-frequency components while suppressing middle and high frequencies. Conversely, for heterophilous datasets, the learned spectral filter of NewtonNet emphasis more on high-frequency components compared to other models.

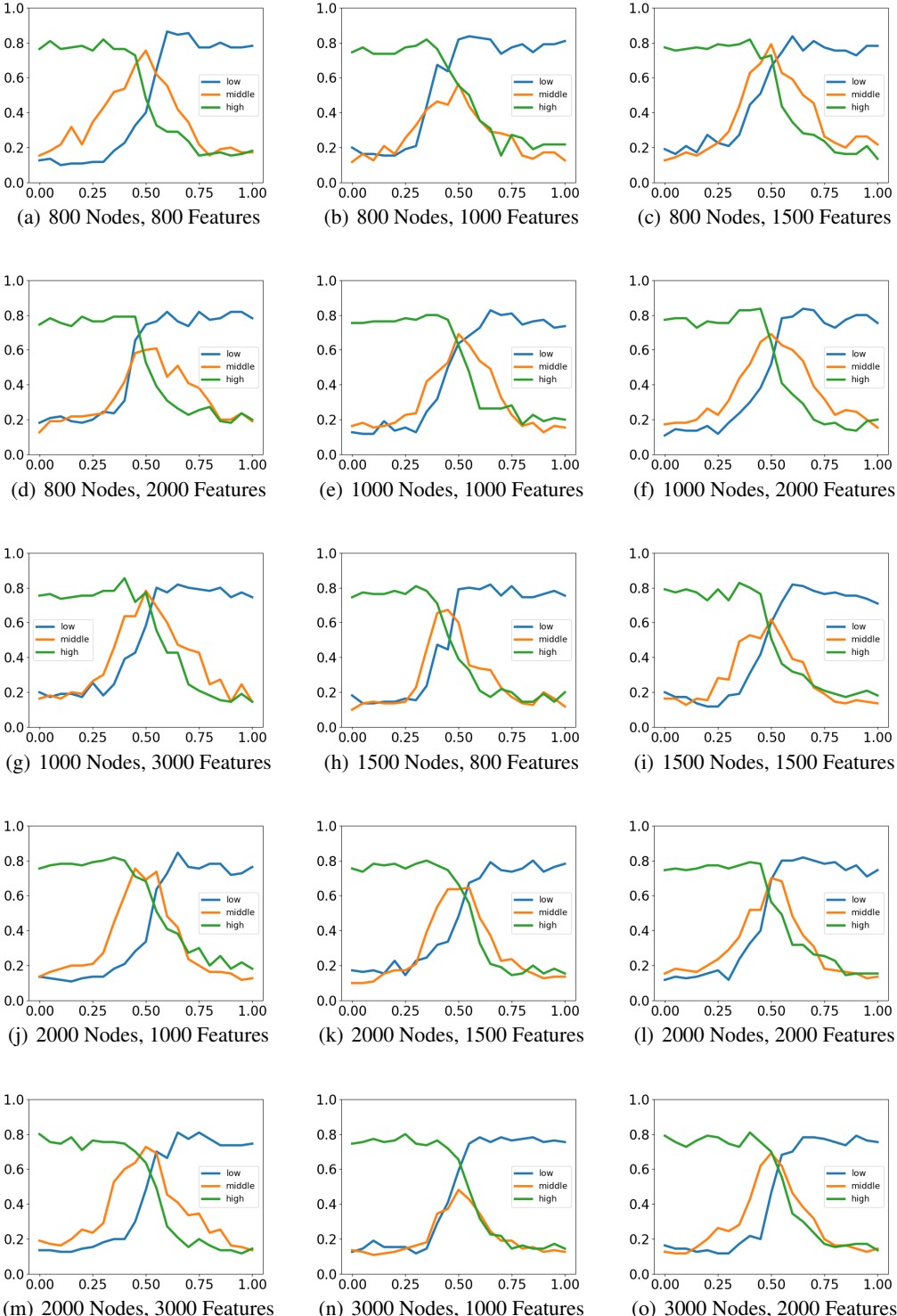

Figure 9: Frequency importance on CSBM model with different hyperparameters.

Table 6: The real homophily ratio and learned homophily ratio in Table 1

|          | Cora | Cite. | Pubm. | Cham. | Squi. | Croc. | Texas | Corn. | Penn. |
|----------|------|-------|-------|-------|-------|-------|-------|-------|-------|
| **Real**    | 0.81 | 0.74 | 0.80 | 0.24 | 0.22 | 0.25 | 0.11 | 0.20 | 0.47 |
| **Learned** | 0.83 | 0.79 | 0.83 | 0.27 | 0.23 | 0.28 | 0.12 | 0.33 | 0.51 |

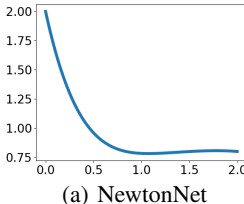
(a) NewtonNet

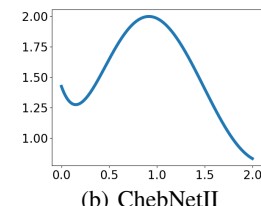
(b) ChebNetII

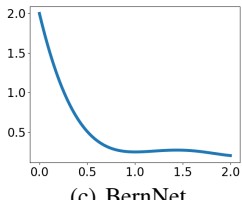
(c) BernNet

Figure 10: Learned filters on Cora.

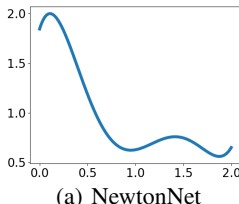
(a) NewtonNet

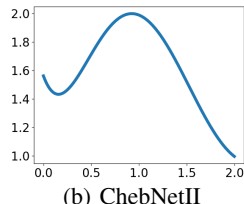
(b) ChebNetII

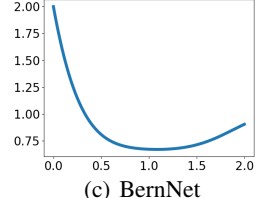
(c) BernNet

Figure 11: Learned filters on Citeseer.

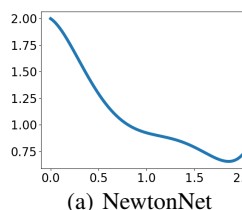
(a) NewtonNet

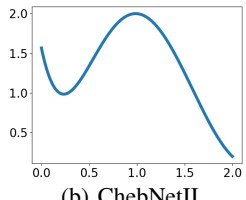
(b) ChebNetII

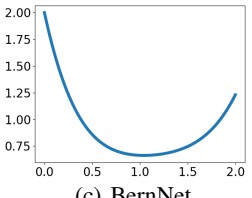
(c) BernNet

Figure 12: Learned filters on Pubmed.

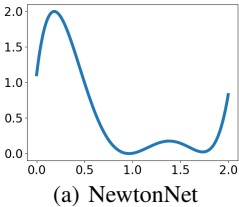
(a) NewtonNet

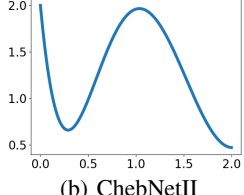
(b) ChebNetII

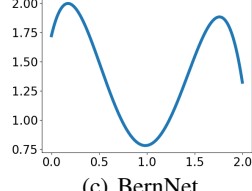
(c) BernNet

Figure 13: Learned filters on Chameleon.

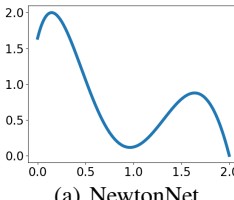 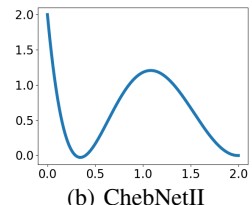 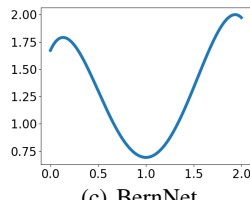

(a) NewtonNet  (b) ChebNetII  (c) BernNet

Figure 14: Learned filters on Squirrel.

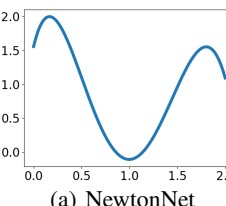 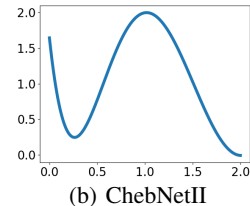 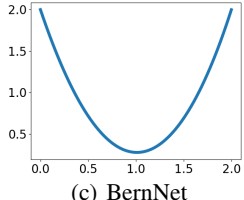

(a) NewtonNet  (b) ChebNetII  (c) BernNet

Figure 15: Learned filters on Crocodile.

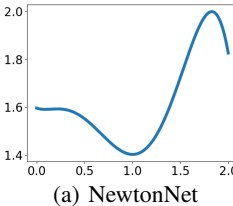 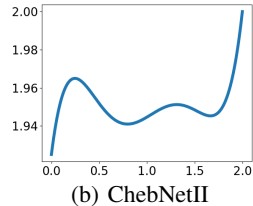 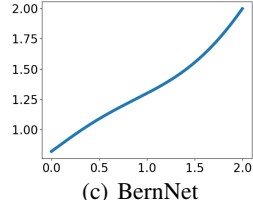

(a) NewtonNet  (b) ChebNetII  (c) BernNet

Figure 16: Learned filters on Texas.

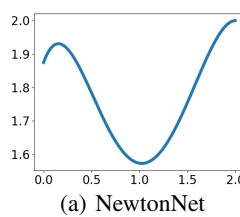 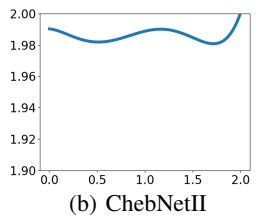 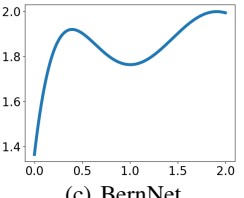

(a) NewtonNet  (b) ChebNetII  (c) BernNet

Figure 17: Learned filters on Cornell.

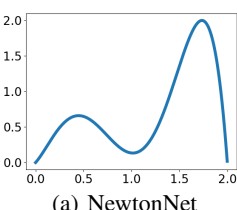 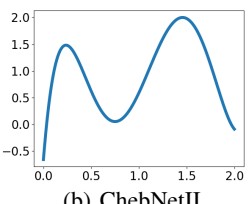 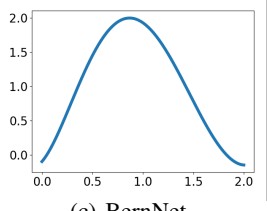

(a) NewtonNet  (b) ChebNetII  (c) BernNet

Figure 18: Learned filters on Penn94.

