# OpenReview forum: "Shape-aware Graph Spectral Learning"
_ICLR.cc/2024/Conference — Submitted to ICLR 2024_

### Official Review · Reviewer_7Vz6 · 2023-10-21

**Soundness:** 3 good
**Presentation:** 3 good
**Contribution:** 3 good
**Rating:** 6
**Confidence:** 3

**Summary:**

This paper studies spectral GNN. Starting from the widely adopted empirical observations that the homophilic level of the interested graph determines which frequencies are preferred in response, the authors present a rigorous theorem to characterize the dependency between the homophilic level and the preferred graph filter. Then, a theoretically motivated novel GNN is proposed, named NewtonNet. To instantiate the idea of encouraging larger amplitude for specific frequencies than the others, NewtonNet is weaponed with interpolation technique rather than approximation, which allows directly regularizing each specific frequency’s response amplitude. The authors also conducted extensive experiments on various node classification tasks with different split ratios. NewtonNet seems to surpass existing methods in most cases. Moreover, a detailed analysis of the learned graph filter implies that NewtonNet’s advantage is rooted in how it aligns with the theorem introduced in this paper, namely, learning low-pass and high-pass filters on homophilic and heterophilic graphs, respectively.

**Strengths:**

1.	This paper is well-written. I can effortlessly pick the core idea(s) up.
2.	The theoretical results are important to this community and are general enough. The importance is due to that a lot of research works have regarded it as a fact, yet such a rigorous analysis is absent. Generality is in the sense that it is not restricted to a specific random graph model, has no unrealistic assumption, and has helpful implications.
3.	The proposed NewtonNet is novel to me. It is not another spectral GNN that just replaces the polynomial family to approximate the desired graph filter. Instead, NewtonNet considers an interpolation technique, which allows it to regularize specific frequencies explicitly. Moreover, no additional computation, especially spectral decomposition, is involved, which ensures NewtonNet is practical and scalable.
4.	The experimental results are convincing, including new heterophilic datasets, and compare different methods at various split ratios.

**Weaknesses:**

1.	In the presented theorem, the two compared graph filters have the same response norm, which is reasonable but not the case for many existing spectral GNNs. My concern is that, although the theorem is correct, some existing spectral GNNs can still fit the desired shape by further increasing the overall norms of their responses. This is not a technical flaw but needs to be further discussed.
2.	In the experimental setup, GCN is regarded as a spatial GNN method, which is not that necessary, in my opinion. To my knowledge, GCN is often analyzed as a simplification of Chebyshev polynomial approximator.

**Questions:**

I am wondering how you calculate the homophilic level ($h$) when there is just a tiny fraction of nodes are labeled. Actually, this is a crucial factor for the explicit shape-aware regularization to work well, according to my understanding of this work.


Update:

I tried your code. There are some questions that need to be answered so that I can reproduce your reported results:

1. How do you set `weight_decay`? It seems that there are typos in D.4, where `dropout` and `dprate` are taken from $ \{ 0, 0.0005 \} $. However, I guess they are chosen from $\{0, 0.5\}$, and `weight_decay` is chosen from $\{0, 0.0005\}$. Am I wrong?
2. I am curious about how you conduct HPO. The search space is supposed to be:

```Python
 lr = trial.suggest_categorical("lr", [0.005, 0.01, 0.05])
 temp_lr = trial.suggest_categorical("temp_lr", [0.005, 0.01, 0.05])
 dropout = trial.suggest_categorical("dropout", [0, 0.5])
 dprate = trial.suggest_categorical("dprate", [0, 0.5])
 wd = trial.suggest_categorical("wd", [0.0, 5e-4])
 gamma1 = trial.suggest_categorical("gamma1", [0, 1, 3, 5])
 gamma2 = trial.suggest_categorical("gamma2", [0, 1, 3, 5])
gamma3 = trial.suggest_categorical("gamma3", [0, 1, 3, 5])
```

It seems that it is intractable to enumerate all possible choices. Instead, I run your code with an HPO toolkit (optuna) with 400 trials, where each trial returns the mean best valid accuracy as the feedback. Other hyper-parameters are kept unchanged (i.e., using your default value `K=5`, `L=2`, and `hidden=64`. The best trial on Cora, CiteSeer, Crocodile, and Gamer are 90.610, 78.797, 76.002, and 61.861, respectively. Obviously, it is not that promising to achieve the reported test accuracy for such configurations, and, as expected, the test accuracy on them are 87.948, 77.293, 74.037, and 61.712, respectively.

---

> ### Author Response · Authors · 2023-11-15
> **Response (1/2)**
>
> Dear Reviewer 7Vz6,
>
> We are grateful to the reviewers for his/her time and thoughtful evaluation of our paper. Here are our responses and answers.
>
> &nbsp;
>
> > W1: In the presented theorem, the two compared graph filters have the same response norm, which is reasonable but not the case for many existing spectral GNNs. My concern is that, although the theorem is correct, some existing spectral GNNs can still fit the desired shape by further increasing the overall norms of their responses. This is not a technical flaw but needs to be further discussed.
>
> **RW1:**
> 1. In Theorem 1, the reason **why we assume two filters have the same response norm is to avoid the trivial solution**. To be more specific, we made two assumptions. (1) before a certain point $m$, $g_1(\lambda_i)<g_2\left(\lambda_i\right)$; after the point $m$, $g_1\left(\lambda_i\right) > g_2\left(\lambda_i\right)$. (2) $g_1$ and $g_2$ have the same output norm.
> If we don’t assume they have the same norm, consider the following scenario: $g_1(\lambda_1)  = g_2(\lambda_1) - e < g_2(\lambda_1) $, and $g_1(\lambda_i)  = g_2(\lambda_i) + E >> g_2(\lambda_i) $, for $i \ge 2$, where $e$ is a very small value ($10^{-5}$) and $E$ is a very large value ($10^{5}$). In this case, assumption 1 is still satisfied. However, because $ E$ is an extremely large value, almost all of the frequencies are involved in the final representations, which are meaningless and contain no information. Therefore, we only compare two filters with the same norm.
>
> 2. **Increasing the overall norm of the output values does not mean that the filter can fit better** because what really matters is the relative difference between high and low frequencies, not their absolute values. Consider we have a certain filter $g$, and then we multiply its output values by 2 to get $g’ = 2g$. Obviously, $||g’||_2^2 = 4||g||_2^2$. However, they have the same expressive power because the output results are just multiplied by a constant.
>
> 3. We adopted these assumptions solely for the purpose of understanding which types of filters are favored by a graph with a specific homophily ratio. However, **in practical applications, we do not impose any restrictions on the norm of the spectral filter**. A filter that incorporates beneficial frequencies and discards harmful frequencies is likely to yield good results. This is proved by our experiments.
>
> &nbsp;
>
> > W2: In the experimental setup, GCN is regarded as a spatial GNN method, which is not that necessary, in my opinion. To my knowledge, GCN is often analyzed as a simplification of the Chebyshev polynomial approximator.
>
> **RW2:**
> Thanks for pointing this out! Yes, you are totally correct. GCN can be considered as both spatial and spectral methods. Actually, according to [1], most GNNs can be explained as both spatial and spectral. And this does not affect our analysis and results.

---

> ### Author Response · Authors · 2023-11-15
> **Response (2/2)**
>
> >Q1: I am wondering how you calculate the homophilic level ($h$) when there is just a tiny fraction of nodes are labeled. Actually, this is a crucial factor for the explicit shape-aware regularization to work well, according to my understanding of this work.
>
> **RQ1:** In the initial epoch, we compute the homophily ratio using the given training labels. Then, **in every subsequent epoch, we update this ratio by considering both the original and the predicted labels.** Table 6 compares the actual homophily ratios with estimated homophily ratios by our method. The similarity between these two sets of ratios indicates that our method can estimate the homophily ratio accurately.
>
> &nbsp;
>
> >Q2: I tried your code. There are some questions that need to be answered so that I can reproduce your reported results...
>
> **RQ2:**
> 1. Yes, you are correct, the weight_decay is chosen from {0, 0.0005}. And we tune the dropout and dprate among {0, 0.1, 0.3, 0.5, 0.7, 0.9}. If you use Optuna, the code would be.
>
> `dropout = trial.suggest_categorical("dropout", [0.0, 0.1, 0.3, 0.5, 0.7, 0.9]) `
> `dprate = trial.suggest_categorical("dprate", [0.0, 0.1, 0.3, 0.5, 0.7, 0.9])`
>
> We have revised the paper accordingly and submitted the new version.
>
>
> 2. Thanks for noticing this. Except that, the "dropout" and "dprate" should also be tuned, other parts are correct. You may increase the number of trials to get better results. **To ensure better reproductivity,** we have updated the code and done the following things:
> (1) We provide our virtual environment and optimal hyperparameters in scripts.sh to reproduce results in Table 1.
> (2) We give the code to search optimal hyperparameters using Optuna.
> (3) We provide the code to reproduce the results in the case study in Section 3.
> All of these are in the instructions in README.md (https://anonymous.4open.science/r/NewtonNet-8115/README.md).
>
> &nbsp;
> Hope this will help! Shall you have any other questions, please feel free to let us know, we will be happy to answer. If there’s no further concern, we sincerely hope that you will increase your rating.
>
>
> &nbsp;
> &nbsp;
>
> Reference:
> [1] Revisiting Graph Neural Networks: All We Have is Low-Pass Filters.

---

> > ### Comment · Reviewer_7Vz6 · 2023-11-15
> > **Discussion**
> >
> > Thanks for your updates. I will try your updated code to reproduce the results. I promise to raise the score according to the results.

---

> > ### Comment · Reviewer_7Vz6 · 2023-11-17
> > **Discussion on Reproducing the Results**
> >
> > Thanks again for updating the code. However, I tried `bash scripts.sh` with
> >
> > ```Bash
> > CUDA_VISIBLE_DEVICES="4" python main.py --dataname cora --lr 0.05 --temp_lr 0.01 --K 5 --hidden 64 --weight_decay 0.0005 --dropout 0.3 --dprate 0.5 --gamma 0 --gamma2 1 --gamma3 3 --save_results >result/out 2>result/err &
> >
> > CUDA_VISIBLE_DEVICES="5" python main.py --dataname citeseer --lr 0.01 --temp_lr 0.05 --K 5 --hidden 64 --weight_decay 0.0 --dropout 0.0 --dprate 0.5 --gamma 0 --gamma2 3 --gamma3 5 --save_results >/dev/null 2>&1 &
> >
> > #python main.py --dataname pubmed --lr 0.05 --temp_lr 0.005 --K 5 --hidden 64 --weight_decay 0.0 --dropout 0.3 --dprate 0.3 --gamma 0 --gamma2 0 --gamma3 1
> >
> > #python main.py --dataname chameleon --lr 0.05 --temp_lr 0.01 --K 5 --hidden 64 --weight_decay 0.0 --dropout 0.0 --dprate 0.0 --gamma 3 --gamma2 5 --gamma3 0
> >
> > #python main.py --dataname squirrel --lr 0.05 --temp_lr 0.005 --K 5 --hidden 64 --weight_decay 0.0 --dropout 0.1 --dprate 0.5 --gamma 5 --gamma2 3 --gamma3 0
> >
> > CUDA_VISIBLE_DEVICES="6" python main.py --dataname crocodile --lr 0.005 --temp_lr 0.005 --K 5 --hidden 64 --weight_decay 0.0005 --dropout 0.0 --dprate 0.1 --gamma 3 --gamma2 3 --gamma3 0 --save_results >/dev/null 2>&1 &
> >
> > #python main.py --dataname texas --lr 0.05 --temp_lr 0.005 --K 5 --hidden 64 --weight_decay 0.0005 --dropout 0.5 --dprate 0.0 --gamma 0 --gamma2 5 --gamma3 3
> >
> > #python main.py --dataname cornell --lr 0.05 --temp_lr 0.005 --K 5 --hidden 64 --weight_decay 0.0005 --dropout 0.5 --dprate 0.1 --gamma 0 --gamma2 3 --gamma3 0
> >
> > #python main.py --dataname Penn94 --lr 0.005 --temp_lr 0.01 --K 5 --hidden 64 --weight_decay 0.0005 --dropout 0.3 --dprate 0.1 --gamma 0 --gamma2 1 --gamma3 0
> >
> > CUDA_VISIBLE_DEVICES="7" python train_binary.py --dataname gamer --lr 0.005 --temp_lr 0.005 --K 5 --hidden 64 --weight_decay 0.0 --dropout 0.5 --dprate 0.3 --gamma 1 --gamma2 0 --gamma3 0 --save_results >/dev/null 2>&1 &
> >
> > #python train_binary.py --dataname genius --lr 0.05 --temp_lr 0.005 --K 5 --hidden 64 --weight_decay 0.0 --dropout 0.0 --dprate 0.5 --gamma 0 --gamma2 1 --gamma3 0
> > ```
> >
> > and got:
> >
> > ```
> > 87.616,89.464
> > 91.312,86.506
> > 89.094,89.649
> > 89.464,85.028
> > 89.649,90.388
> > 89.427,88.207
> > ```
> >
> > ```
> > 79.398,80.602
> > 76.842,74.737
> > 76.992,75.038
> > 78.195,76.541
> > 77.444,77.895
> > 77.774,76.962
> > ```
> >
> > ```
> > 79.235,74.635
> > 76.870,77.171
> > 76.741,76.569
> > 77.773,76.311
> > 77.343,76.827
> > 77.592,76.303
> > ```
> >
> > ```
> > 60.717,60.264
> > 61.655,61.512
> > 60.935,60.977
> > 62.286,62.213
> > 61.837,62.307
> > 61.486,61.454
> > ```
> >
> > on Cora, CiteSeer, Crocodile, and Gamer, respectively.
> >
> > It seems that these results are inconsistent with those reported in your submission. Could you teach me how to produce the exact results? Thanks!

---

> > > ### Author Response · Authors · 2023-11-17
> > >
> > > Thanks for your reply.
> > >
> > > 1. Note that different environments may lead to difference in results. Did you use the same environment as `requirements.txt`? Make sure you created the same virtual environment according to the `README.md` and use the same random seed.
> > >
> > > 2. Different machines may also cause different results. We run our experiments on NVIDIA RTX A6000. Could you provide more details of your environments and machines?
> > >
> > > 3. To get results on a specific dataset, you don't need to run `bash scripts.sh`.
> > >
> > > For example, if you want to get results on Cora, you can run
> > > `python main.py --dataname cora --lr 0.05 --temp_lr 0.01 --K 5 --hidden 64 --weight_decay 0.0005 --dropout 0.3 --dprate 0.5 --gamma 0 --gamma2 1 --gamma3 3`
> > >
> > >
> > > To get results on Crocodile, run
> > > `python main.py --dataname crocodile --lr 0.005 --temp_lr 0.005 --K 5 --hidden 64 --weight_decay 0.0005 --dropout 0.0 --dprate 0.1 --gamma 3 --gamma2 3 --gamma3 0`
> > >
> > > 4. You can search your own optimal hyperparameters with
> > > ```
> > > python search_multi.py --dataname $DATASET
> > > python search_binary.py --dataname $DATASET
> > > ```
> > > , where `$DATASET` is the name of the dataset.
> > >
> > >
> > > 5. We have noticed that some of your results are better than our reported results, while others are not as good as ours. We believe this may be caused by randomness across different environments and machines.
> > >
> > > Since figures cannot be posted on Openreview, we would be more than happy to discuss further details after the code becomes publicly available.

---

> > > > ### Comment · Reviewer_7Vz6 · 2023-11-21
> > > > **Discussion**
> > > >
> > > > Thanks for your explanation. I like the idea of interpolation and tend to accept your submission. However, I am opposed to your point about the randomness in spectral GNN's evaluation. If it is the case, then what researchers have done in this line of research would be meaningless in my opinion. Anyway, I strongly suggest you to carefully prepare your code repository and release it later.

---

> > > > > ### Author Response · Authors · 2023-11-23
> > > > >
> > > > > Thank you for your valuable support and insightful suggestions. We will carefully prepare and publish the code upon acceptance.

---

### Official Review · Reviewer_LpUg · 2023-10-30

**Soundness:** 4 excellent
**Presentation:** 4 excellent
**Contribution:** 3 good
**Rating:** 6
**Confidence:** 4

**Summary:**

This work proposes a novel approach to spectral graph neural networks called Shape-Aware Graph Spectral Learning. The authors introduce a regularization term that incorporates prior knowledge about the desired shape of the corresponding homophily level, which improves the performance of the proposed NewtonNet on both homophilous and heterophilous datasets. The paper also discusses the limitations of message-passing GNNs and how spectral GNNs overcome them. Overall, the paper presents a new perspective on spectral GNNs and provides insights into how to improve their performance.

**Strengths:**

> A substantive assessment of the strengths of the paper, touching on each of the following dimensions: originality, quality, clarity, and signicance. We encourage reviewers to be broad in their denitions of originality and signicance. For example, originality may arise from a new denition or problem formulation, creative combinations of existing ideas, application to a new domain, or removing limitations from prior results. You can incorporate Markdown and Latex into your review. See /faq (/faq).

1. Proposes a novel approach to spectral graph neural networks that takes into account the relationship between graph frequency and homophily/heterophily.
2. Incorporates prior knowledge about the desired shape of the corresponding homophily level, which improves the performance of the proposed NewtonNet on both homophilous and heterophilous datasets.
3. Provides a detailed analysis of the proposed approach and its performance on various datasets, which adds to the understanding of spectral GNNs.
4. Discusses the limitations of message-passing GNNs and how spectral GNNs overcome them, which provides insights into the strengths and weaknesses of different GNN architectures.

**Weaknesses:**

> A substantive assessment of the weaknesses of the paper. Focus on constructive and actionable insights on how the work could improve towards its stated goals. Be specic, avoid generic remarks. For example, if you believe the contribution lacks novelty, provide references and an explanation as evidence; if you believe experiments are insucient, explain why and exactly what is missing, etc.

1. The effectiveness of the suggested approach might not be universally applicable, as its performance is contingent on the specific characteristics of the dataset, making it less versatile.

2. The process of fine-tuning hyperparameters could be a resource-intensive and time-consuming endeavor, potentially impeding the scalability of the proposed approach.

3. The foundation of our approach hinges on the presumption that the graph Laplacian matrix adequately encapsulates the graph's inherent structure; however, this assumption may not hold true in all instances.

4. Interpretability poses a concern, as the proposed approach lacks a straightforward means of explanation. This could curtail its utility in domains where interpretability is a paramount consideration.

5. The susceptibility of the proposed approach to data noise and outliers may undermine its performance when confronted with real-world datasets.

6. A significant limitation arises from the substantial volume of labeled data that the proposed approach necessitates to attain satisfactory performance, which can be impractical in situations where acquiring labeled data is a challenging and expensive endeavor.

**Questions:**

> Please list up and carefully describe any questions and suggestions for the authors. Think of the things where a response from the author can change your opinion, clarify a confusion or address a limitation. This is important for a productive rebuttal and discussion phase with the authors.

1. Could the proposed approach be adapted to accommodate directed graphs? While the paper focuses on undirected graphs, considering the prevalence of directed graphs in real-world applications, it would be intriguing to explore the adaptability of this approach to handle directed graphs.
2. How does the performance of the proposed approach compare to other state-of-the-art methods on the same datasets? The paper presents comparisons with baseline methods, but a comprehensive evaluation against other contemporary approaches on the same datasets would offer valuable insights into its relative effectiveness.
3. Could the authors offer additional insights into the hyperparameter tuning process? While the paper briefly mentions hyperparameter tuning, more extensive details on the selection process and an assessment of the sensitivity of results to hyperparameter choices would enhance clarity.
4. What is the scalability of the proposed approach to larger graphs? The current evaluation primarily focuses on relatively small graphs. Understanding its performance and computational complexity as graph size scales would be beneficial, particularly for applications involving larger graphs.
5. Can the authors provide a deeper understanding of the interpretability challenges associated with the proposed approach? The paper notes potential interpretability issues, but additional insights into the specific challenges and potential pathways to improve interpretability would be valuable.
6. How does the proposed approach handle noisy or incomplete data? The paper does not explicitly address the approach's handling of noisy or incomplete data, which can be a limitation in real-world scenarios. It would be advantageous to gain insights into potential adaptations of the proposed approach to accommodate such data scenarios.

---

> ### Author Response · Authors · 2023-11-15
> **Response (1/2)**
>
> Dear Reviewer LpUg,
>
> Thank you for your dedication to reviewing our paper. We would like to kindly remind the reviewer that **many of the questions/comments are irrelevant to our paper.** For example, your comments state interpretability, outliers, and noisy data in Weaknesses 4-5 and Questions 5-6, which have nothing to do with our paper. Our paper works on graph spectral learning, rather than the above areas.
> **There are also paradoxical sentences and expressions.** For example, Question 2 both acknowledges and denies our comparison with other SOTA baselines.
> **The remaining comments/questions are general for most graph mining papers.** For example, scalability, and comparison with other baselines. We highly suspect that you submitted a wrong review of our paper. We sincerely hope that you could double-check our paper and your review. Nevertheless, the following are our responses to your comments and questions:
>
> &nbsp;
>
> >W1: The effectiveness of the suggested approach might not be universally applicable, as its performance is contingent on the specific characteristics of the dataset, making it less versatile.
>
> **RW1:** In Table 1, we demonstrate that our method achieves state-of-the-art performance across 11 diverse datasets. These datasets span various domains (Citation, WebKB, Wikipedia, Social networks) and range from small to large scales. We believe this variety and the consistent performance across these datasets substantiate the effectiveness of our method beyond mere coincidence.
>
>
> >W2: The process of fine-tuning hyperparameters could be a resource-intensive and time-consuming endeavor, potentially impeding the scalability of the proposed approach.
>
> **RW2:**
> 1. We detail the experimental settings and the process for hyperparameter tuning in Section 6.1 and Appendix D.4. Once the search space is determined, methods like grid search or hyperparameter optimization tools such as Optuna can be utilized to identify the optimal hyperparameters.
> 2. The time and space complexity of NewtonNet is analyzed in Appendix B.1, where we demonstrate the excellent scalability of our method. NewtonNet has a better scalability than GCN.
>
>
> > W3: The foundation of our approach hinges on the presumption that the graph Laplacian matrix adequately encapsulates the graph's inherent structure; however, this assumption may not hold true in all instances.
>
> **RW3:** As we discuss in Section 2, the graph Laplacian matrix $\mathbf{L}=\mathbf{I}-\mathbf{D}^{-1 / 2} \mathbf{A} \mathbf{D}^{-1 / 2}$ encapsulates all information from the adjacency matrix $\mathbf{A}$. This is the same across all graphs.
>
>
> > W4: Interpretability poses a concern, as the proposed approach lacks a straightforward means of explanation. This could curtail its utility in domains where interpretability is a paramount consideration.
>
> **RW4:** The focus of our paper is not on interpretability, but rather on exploring the relationship between homophily/heterophily and graph spectral learning.
>
>
> >W5: The susceptibility of the proposed approach to data noise and outliers may undermine its performance when confronted with real-world datasets.
>
> **RW5:** As mentioned in response to RW1, our experiments are conducted on 11 varied real-world datasets. Our research does not specifically address noisy data or outlier detection; instead, it concentrates on examining the relationship between homophily/heterophily and graph spectral learning.
>
>
> > W6: A significant limitation arises from the substantial volume of labeled data that the proposed approach necessitates to attain satisfactory performance, which can be impractical in situations where acquiring labeled data is a challenging and expensive endeavor.
>
> **RW6:** Our proposed method does not require a substantial volume of labeled data. The experiments illustrated in Figure 3 reveal that NewtonNet demonstrates a higher percentage of improvements under conditions of limited labels compared to scenarios with higher training ratios. This indicates that NewtonNet performs effectively even with a limited number of labels available.

---

> ### Author Response · Authors · 2023-11-15
> **Response (2/2)**
>
> >Q1: Could the proposed approach be adapted to accommodate directed graphs? While the paper focuses on undirected graphs, considering the prevalence of directed graphs in real-world applications, it would be intriguing to explore the adaptability of this approach to handle directed graphs.
>
> **RQ1:** Currently, the Laplacian matrix definition in all spectral GNNs is based on undirected graphs, as this ensures the matrix is positive semidefinite with eigenvalues between [0, 2]. However, we acknowledge the prevalence of directed graphs in many real-world scenarios and are keen to investigate this problem in our future work.
>
> > Q2: How does the performance of the proposed approach compare to other state-of-the-art methods on the same datasets? The paper presents comparisons with baseline methods, but a comprehensive evaluation against other contemporary approaches on the same datasets would offer valuable insights into its relative effectiveness.
>
> **RQ2:** In Table 1, we compare our NewtonNet with 10 other representative and state-of-the-art methods, encompassing both spatial (such as MLP, Mixhop, GCN, APPNP, GloGNN++) and spectral (like ChebNet, GPRGNN, ChebNetII, BernNet, JacobiConv) approaches. We believe this comprehensive comparison strongly validates the effectiveness of our method.
>
> >Q3: Could the authors offer additional insights into the hyperparameter tuning process? While the paper briefly mentions hyperparameter tuning, more extensive details on the selection process and an assessment of the sensitivity of results to hyperparameter choices would enhance clarity.
>
> **RQ3:**
> 1. Yes, we determined the optimal hyperparameters through grid search, detailed in Section 6.1 and Appendix D.4. Hyperparameter optimization tools like Optuna can also be employed for this purpose.
> 2. Sensitivity analysis conducted in Appendix E.1 examines the impact of parameters $\gamma_1, \gamma_2, \gamma_3$, and $K$ on our model.
>
> >Q4: What is the scalability of the proposed approach to larger graphs? The current evaluation primarily focuses on relatively small graphs. Understanding its performance and computational complexity as graph size scales would be beneficial, particularly for applications involving larger graphs.
>
> **RQ4:**
> 1. The time and space complexity of NewtonNet, analyzed in Appendix B.1, underscores our method's superior scalability, outperforming GCN in this regard.
> 2. Our experiments included several large-scale datasets. For instance, Penn94, Twitch-gamer, and Genius[1] have 41554, 421961, and 168114 nodes, respectively. Table 5 provides detailed statistics of these datasets.
>
> >Q5: Can the authors provide a deeper understanding of the interpretability challenges associated with the proposed approach? The paper notes potential interpretability issues, but additional insights into the specific challenges and potential pathways to improve interpretability would be valuable.
>
> **RQ5:** Please refer to RW4.
>
> >Q6: How does the proposed approach handle noisy or incomplete data? The paper does not explicitly address the approach's handling of noisy or incomplete data, which can be a limitation in real-world scenarios. It would be advantageous to gain insights into potential adaptations of the proposed approach to accommodate such data scenarios.
>
> **RQ6:** Please refer to RW5.
>
> &nbsp;
> We hope these responses clarify your queries. Should you have any further questions, please do not hesitate to let us know.
>
> &nbsp;
>
> Reference:
> [1] Large Scale Learning on Non-Homophilous Graphs: New Benchmarks and Strong Simple Methods

---

> ### Author Response · Authors · 2023-11-23
>
> Thank you for your contributions and feedback. As the author-reviewer discussion period is ending soon, we'd appreciate it if you could re-evaluate our submission with the new experiments and clarifications in mind. If you have additional insights, we're open to continued discussion until the deadline.

---

> > ### Comment · Reviewer_LpUg · 2023-12-01
> >
> > Thanks for your comprehensive responses addressing the hyperparameters and scalability concerns. Your clear and persuasive explanations regarding the weaknesses have significantly improved my understanding.
> > I'm inclined to reconsider and potentially raise my rating to a 6 based on your thorough and convincing clarifications.

---

### Official Review · Reviewer_zWto · 2023-11-01

**Soundness:** 3 good
**Presentation:** 3 good
**Contribution:** 3 good
**Rating:** 6
**Confidence:** 4

**Summary:**

This paper proposes a novel GNN architecture called NewtonNet that uses Newton interpolation to better grasp the desired shape of learned polynomial filters via regularization. The prior knowledge of the desired shape comes from well-built theoretical analyses of the relationships between homophily ratio and low- or high-frequency importance. The proposed method achieves superior or comparable performances compared to other state-of-the-art methods.

**Strengths:**

S1. The theoretical part is contributive and justified with well-designed experiments.

S2. The idea of controlling the shape of a polynomial filter via Newton nodes is interesting.

S3. The paper is well-organized.

S4. The code is accessible.

**Weaknesses:**

> W1. On the choice of K.

First, In the second line under Eq.7, the authors write that they set K=4 in this paper. According to Fig.2  and the description of experimental settings in Appendix D.4, you want to write K=5, right?

Then, in Appendix E.1, the authors conduct a sensitivity analysis on K, and find that NewtonNet's performances on the Cora and Chameleon datasets peak at K=5. Such an analysis is weird since **the accuracies on test datasets are in fact used as prior knowledge**. A valid way is to set K as a hyperparameter.

I generally like this work, but have to discuss further on this issue with the chair.

> W2. On the weak-supervised experiments.

According to Ref.1, the Chameleon and Squirrel datasets are problematic in that they use duplicated nodes. This would be problematic, especially in weak-supervised settings. I am curious how this experiment would perform on other datasets.



Ref1. Platonov O, Kuznedelev D, Diskin M, et al. A critical look at the evaluation of GNNs under heterophily: are we really making progress?

**Questions:**

Q1. Can the regularization be used for other polynomial filtering functions?

Q2. The authors explain in Appendix E.1 that the reason for their choice of K to be 5 (typically set as 10 or tuned in a range in other polynomial filters) is that when K is larger, the polynomial becomes susceptible to the Runge phenomenon according to He et al., (2022).  Since the Runge phenomenon is caused by the selection of equal-paced interpolation points, why not use Chebyshev nodes for interpolation?

---

> ### Author Response · Authors · 2023-11-15
> **Response (1/2)**
>
> Dear Reviewer zWto,
>
> Thank you for your comments on improving our work. Here are our responses and answers.
>
> &nbsp;
>
> > W1. On the choice of K.
> First, In the second line under Eq.7, the authors write that they set K=4 in this paper. According to Fig.2 and the description of experimental settings in Appendix D.4, you want to write K=5, right? Then, in Appendix E.1, the authors conduct a sensitivity analysis on K, and find that NewtonNet's performances on the Cora and Chameleon datasets peak at K=5. Such an analysis is weird since the accuracies on test datasets are in fact used as prior knowledge. A valid way is to set K as a hyperparameter.
>
> **RW1:**
> 1. Thanks for your careful reading. Yes, it’s a typo. We have revised the paper accordingly and submitted the new version.
> 2. We appreciate your question and realize our explanation might not have been as clear as intended:
> (1) We assure you that **we did not use any prior knowledge from the test sets in our experiments**. We strictly follow the common practice of training the model on the training set, selecting the best model (hyperparameters) based on the validation set, and then evaluating and reporting the results on the test set.
> (2) **$K$ is indeed a hyperparameter.** Regarding the hyperparameter $K$,  to simplify our approach and reduce the search space, we initially did not tune $K$ and chose $K=5$ for the results presented in Table 1. However, to analyze the impact of $K$, we varied it over the set {2, 5, 8, 11, 14} as shown in Fig. 8. This investigation revealed that $K=5$ yields the best outcomes for the Cora and Chameleon datasets. However, it doesn’t mean that $K=5$ is the best choice for other datasets. Additionally, we conducted further experiments on amazon-ratings, which shows that $K=8$ provides better results. Generally, as $K$ increases from a small number the performance will first increase and then drop when $K$ is too large. A number of $K$ between 5 to 8 works well.
>
> |                  | **amazon-ratings** |
> |:----------------:|:------------------:|
> |  NewtonNet (K=2) |      54.43±0.2     |
> |  NewtonNet (K=5) |      55.06±0.5     |
> |  NewtonNet (K=8) |    **55.24±0.4**   |
> | NewtonNet (K=11) |      54.81±0.3     |
> | NewtonNet (K=14) |      53.47±0.6     |
>
> (3) To give more facts, if you refer to our code (Line 13 of https://anonymous.4open.science/r/NewtonNet-8115/config.py), it can be seen that $K$ is set as a hyperparameter.
>
>
> &nbsp;
>
>
> > W2. On the weak-supervised experiments. According to Ref.1, the Chameleon and Squirrel datasets are problematic in that they use duplicated nodes. This would be problematic, especially in weak-supervised settings. I am curious how this experiment would perform on other datasets.
>
> **RW2:** We run experiments on your suggested datasets and show the results here. We will add this table and reference in the camera-ready version.
>
> |           | **roman-empire** | **amazon-ratings** |  **tolokers** |
> |:---------:|:----------------:|:------------------:|:-------------:|
> |    MLP    |     66.33±0.9    |      48.85±0.2     |   75.36±0.4   |
> |    GCN    |     61.96±0.8    |      49.90±0.1     |   77.13±0.4   |
> |   Mixhop  |    _74.38±0.5_   |      51.44±0.6     | **80.62±0.5** |
> |   APPNP   |     73.50±0.7    |     _51.80±0.6_    |   75.02±0.2   |
> |   GPRGNN  |     71.09±0.3    |      47.54±0.5     |   73.83±0.6   |
> | ChebNetII |     70.78±0.5    |      46.24±0.7     |   71.86±0.6   |
> | NewtonNet |   **75.34±0.3**  |    **55.24±0.4**   |  _79.01±0.7_  |

---

> > ### Comment · Reviewer_zWto · 2023-11-16
> > **Re: Response (1/2)**
> >
> > Thank you for your explanation. I'm still a bit puzzled because **although you've defined K as a hyperparameter, it remains consistently set at 5 in Table 1**. I understand your clarification regarding Appendix E.1, indicating that this analysis doesn't influence your decision to set K as 5. However, there is still a discrepancy between the information presented in Table 1 and the experiment's description. Is it possible to replace the results in Table 1 with those obtained by treating K as a hyperparameter? I understand that this might require some extra time, but given that you're utilizing Optuna, I presume the process might not be overly time-consuming.

---

> ### Author Response · Authors · 2023-11-15
> **Response (2/2)**
>
> > Q1. Can the regularization be used for other polynomial filtering functions?
>
> > Q2. The authors explain in Appendix E.1 that the reason for their choice of K to be 5 (typically set as 10 or tuned in a range in other polynomial filters) is that when K is larger, the polynomial becomes susceptible to the Runge phenomenon according to He et al., (2022). Since the Runge phenomenon is caused by the selection of equal-paced interpolation points, why not use Chebyshev nodes for interpolation?
>
>
> **RQ1 & RQ2:** Questions 1 and 2 raise similar issues, so I will address them together.
>
> 1. We would like to kindly remind the reviewer that we discussed this on page 7, under the section “Compared with other approximation and interpolation methods,” and in Appendix C. To make it more clear, **shape-aware regularization is applicable exclusively to interpolation methods, as it necessitates direct control over the output values of interpolated points** (specifically, $\mathbf{t}$ in Equation 7). Only interpolation methods directly reflect the output values of the polynomial. Methods like GPRGNN, ChebNet, and BernNet, which use approximation methods, are not compatible with shape-aware regularization.
>
> 2. **While ChebNetII employs interpolation methods, its interpolated points are fixed**, limiting our ability to tailor any interpolated points to fit various shapes of spectral filters. This limitation led us to propose Newton interpolation for learning the spectral filter, which is not only an interpolation method but also allows for the random selection of interpolated points.
>
> 3. NewtonNet can choose any points as interpolated points, including Chebyshev nodes. However, when conducting experiments, we found that Chebyshev nodes with regularization did not yield as satisfactory results as equispaced points. In fact, the performance was similar to ChebNetII. Consequently, we opted for equispaced points in our experiments.
>
> 4. Additionally, **the concern of Runge's phenomenon primarily arises with high polynomial powers.** According to Wikipedia[1], “Runge's phenomenon is a problem of oscillation at the edges of an interval that occurs when using polynomial interpolation with polynomials of high degree …” Our experiments demonstrate that NewtonNet with K=5 delivers good performance across various datasets. Opting for **a lower polynomial degree not only mitigates the Runge phenomenon but also enhances the speed and efficiency of our method.**
>
> &nbsp;
>
> Please let us know if you have further questions!
>
> &nbsp;
> &nbsp;
>
> Reference:
> 1. https://en.wikipedia.org/wiki/Runge%27s_phenomenon

---

> > ### Comment · Reviewer_zWto · 2023-11-16
> > **Re: Response (2/2)**
> >
> > 1. I mean, can your shape-aware regularization be used for approximation-based filters once you substitute $t_i$ with $f(q_i)$ ?
> > 2. **More critical discussion on Runge's Phenomenon.** Considering that setting K=10 typically does not lead to the Runge's phenomenon or reduced performances in other polynomial-based GNNs,  the following claim in the paper might be overstated:
> >
> > > Conversely, if K is set to a large value, the polynomial becomes susceptible to the Runge phenomenon.
> >
> > I reference Chebyshev interpolation here because choosing Chebyshev nodes mitigates the Runge phenomenon [Ref1], potentially aiding NewtonNet in accommodating higher degrees.  However, as you claimed in your response,
> > > ... when conducting experiments, we found that Chebyshev nodes with regularization did not yield as satisfactory results as equispaced points. In fact, the performance was similar to ChebNetII.
> >
> > Does this fact indicate that Runge's phenomenon might not be the primary reason why K=5 is preferable over higher orders in your case?
> >
> > Ref1. https://en.wikipedia.org/wiki/Runge%27s_phenomenon#Change_of_interpolation_points

---

> ### Author Response · Authors · 2023-11-18
>
> Thanks for your reply.
>
>
> > Q1: Thank you for your explanation. I'm still a bit puzzled because although ... I presume the process might not be overly time-consuming.
>
> A1. We do define $K$ as a hyperparameter. The only thing is that we fixed $K=5$ to reduce the search space and simplify the settings. However, according to your suggestions, we re-ran the experiments with $K$ varying among [2, 5, 8, 11, 14] and found some improved results. Due to limited time, we only conducted experiments on Cora, Citeseer, Pubmed, Chameleon, Squirrel, and Crocodile. The results show that on Cora, Citeseer, and Chameleon, the best results are still achieved with $K=5$ and are the same as Table 1. However, on Pubmed, Squirrel, and Crocodile, better results were observed since we expanded the search space. These are summarized in the following table.
>
> | **Dataset** | **Acc (%)** | **Optimal $K$** |
> |:-----------:|:-------:|:-------------:|
> |    Pubmed   |  90.28  |       2       |
> |   Squirrel  |  63.37  |       14      |
> |  Crocodile  |  76.80  |       8       |
>
> Due to time constraints, we conducted only a limited number of trials with Optuna. We believe to achieve better results with more trials. We intend to complete all the experiments and update the results in the next version of our work. Thank you for your valuable suggestions.
>
> &nbsp;
>
>
> > Q2: I mean, can your shape-aware regularization be used for approximation-based filters once you substitute $t_i$ with $f(q_i)$ ?
>
> A2. I’m not sure if I understand your question correctly. I guess $f$ here is the function that we want to approximate, i.e. the spectral filter.
> Since there are no interpolated points $q_i$ in approximation-based filters, we cannot apply shape-aware regularization by substituting $f(q_i)$ directly. But we can also define a series of points $q_i$ on approximation methods.
> Then, let us clarify some concepts. In NewtonNet, $t_k$’s are function values, while $\hat{g}_{t}[q_0, \cdots, q_k]$ are coefficients.
>
> Consider an approximation-based filter, for example, the one used in GPRGNN
> $f(q_i) = \sum_{k=0}^K w_k q_i^k$, where $w_k$’s are coefficients and $f(q_i)$’s are function values.
>
> In interpolation methods, our parameters to be learned are the function values. These function values are then utilized to compute the coefficients. We directly regularize and learn these function values.
>
> In contrast, approximation methods are designed to learn coefficients first. These coefficients are used to calculate function values. Then we regularize the function values. This process requires additional computational steps. Moreover, the regularization in this case is not applied directly to the parameters we aim to learn.
>
> In conclusion, based on your description, while shape-aware regularization might be applicable to approximation methods, **its application is relatively indirect and less efficient**. We believe that exploring this further would be a suitable topic for future research.
>
>
> &nbsp;
>
> >Q3: More critical discussion on Runge's Phenomenon.
>
> A3.
> 1. We didn’t mean $K=10$ is a large value that suffers from  Runge's phenomenon. We mean that the reason why we chose $K=5$ is to mitigate Runge's phenomenon and make the method more efficient.
> 2. Figure 8 shows that $K=5$ is preferable by Cora and Chameleon. However, this doesn’t mean “ $K=5$ is preferable over higher orders ” on all datasets. The added experiments above indicate that Pubmed shows better results with $K=2$, while Crocodile performs better with $K=8$ compared to $K=5$.
>
> &nbsp;
> We hope this addresses your questions.

---

> > ### Comment · Reviewer_zWto · 2023-11-19
> > **Soundness score increased**
> >
> > Thanks for your reply. I have increased the soundness score to 3.

---

### Official Review · Reviewer_UTx2 · 2023-11-01

**Soundness:** 3 good
**Presentation:** 3 good
**Contribution:** 2 fair
**Rating:** 6
**Confidence:** 4

**Summary:**

This paper proposes a novel framework for learning graph filters. The novelty of the framework lies in incorporating the prior information about graph homophily/heterophily into the learning procedure. Theoretical results demonstrate that homophily and heterophily are indeed relevant aspects to consider in inference tasks performed by GNNs. Further, experiments demonstrate that the proposed framework outperforms other baselines or achieves performance close to the best baseline on the node classification task.

POST REBUTTAL

I thank the authors for their response. There are clear conceptual gaps that inhibit the complete understanding of how the homophily information dictates quality of learning (for instance, I find the authors' claim *if we do not provide guidance to the spectral filter, it will learn some undesired information on both small and large graphs* to be highly unconvincing as it has no theoretical or empirical support). Therefore, I have decreased the soundness score to 3 and contribution score to 2, while retaining my original recommendation.

**Strengths:**

The paper is well-written and provides the relevant message of the importance of homophily in learning of spectral graph filters coherently. Theoretical statements with elementary numerical analysis in Section 3 is well done.

**Weaknesses:**

In practice, incorporating homophily information might offer the most significant benefits when the datasets are of limited size. For sufficiently large datasets, the graph filters will be fine-tuned automatically according to the graph homophily level.

**Questions:**

1. I recommend that the complexity analysis in Appendix B.1 be included in the main body of the paper.

2. Are the high frequencies completely irrelevant for graphs with homophily and vice-versa? Currently, the paper discusses relative importance of high and low frequencies for different homophily levels. However, it is not clear to me whether certain frequencies can be completely ignored in the extremities of graph homophily level.

---

> ### Author Response · Authors · 2023-11-15
>
> Dear Reviewer UTx2,
>
> Thank you for your valuable comments on our work. Below are our responses.
> &nbsp;
>
> >W1: In practice, incorporating homophily information might offer the most significant benefits when the datasets are of limited size. For sufficiently large datasets, the graph filters will be fine-tuned automatically according to the graph homophily level.
>
> **RW1:**
> 1. Our approach does not explicitly incorporate homophily information. We rely solely on the graph structure and predicted labels to estimate the homophily ratio.
> 2. Could you kindly point out that is there any reference for the claim “For sufficiently large datasets, the graph filters will be fine-tuned automatically according to the graph homophily level”? To our understanding, there is no clear evidence that graph filters adapt more effectively to larger graphs. Even if the graph is large, if there are not sufficient labels, we still do not know much about the homophily/heterophily information.  In such scenarios, the graph filter's fine-tuning is hindered by insufficient information. Our experiments, illustrated in Figure 3, demonstrate that under conditions of scarce labels, NewtonNet shows a more significant improvement compared to situations with higher training ratios on both large and small datasets. For instance, in the Chameleon dataset, NewtonNet shows a notable improvement over BernNet by 28.2% when the training ratio is 0.1; however, this improvement is 3.2% when the training ratio is 0.6.
> However, **if we do not provide guidance to the spectral filter, it will learn some undesired information on both small and large graphs.** Actually, letting the graph filter be fine-tuned automatically on any graph is exactly what we aim to achieve by proposing shape-aware regularization. The experiments in Table 1 and Figure 3 validate that NewtonNet successfully learns beneficial frequencies autonomously on both small and large datasets, supporting our approach.
>
> &nbsp;
>
>
> > Q1: I recommend that the complexity analysis in Appendix B.1 be included in the main body of the paper.
>
> **RQ1:** Thanks for your suggestion. In the new version submitted, we added a complexity analysis part in Section 4.
>
> &nbsp;
>
>
> >Q2: Are the high frequencies completely irrelevant for graphs with homophily and vice-versa? Currently, the paper discusses the relative importance of high and low frequencies for different homophily levels. However, it is not clear to me whether certain frequencies can be completely ignored in the extremities of graph homophily level.
>
> **RQ2:**
> 1. We explore the relationship between the homophily ratio and frequency importance in Fig. 1(b). This figure demonstrates that even in extreme cases (homophily ratio = 0.0 or 1.0), the importance of low, middle, and high frequencies is not negligible. Therefore, **we cannot conclusively state that “high frequencies are completely irrelevant for entirely homophilous graphs or the reverse.”**
>
> 2. In fact, it's **not necessary to correlate specific homophily ratios with particular frequencies**. NewtonNet, enhanced by shape-aware regularization, is designed to automatically learn the significance of each frequency and adapt to the optimal spectral filter.
>
> 3. In completely homophilous graphs (homophily ratio = 1.0), the graph essentially reduces to several disconnected components/subgraphs, with each subgraph having nodes of the same label. In such scenarios, the node classification task becomes redundant as the class of each node can be easily determined.
>
> &nbsp;
>
> Should you have any further questions or require additional information, please feel free to contact us. If we have addressed your concerns, we sincerely hope you can raise your score.

---

> > ### Author Response · Authors · 2023-11-23
> >
> > Thank you for your contributions and feedback. As the author-reviewer discussion period is ending soon, we'd appreciate it if you could re-evaluate our submission with the new experiments and clarifications in mind. If you have additional insights, we're open to continued discussion until the deadline.

---

### Author Response · Authors · 2023-11-15

We thank all reviewers for their comprehensive evaluations and insightful opinions. We appreciate that our contributions have received positive recognition, which includes:

**Important Research Topic:** The paper tackles an essential topic in Graph Neural Networks, focusing on the impact of graph homophily and heterophily in learning graph spectral filters. (Reviewer UTx2)

**Solid Theoretical Analysis:** The paper gives a solid theoretical foundation about the relationship between spectral filters and graph homophily/heterophily, highlighting its relevance and generality in the field of GNNs. (Reviewers zWto and 7Vz6)

**Novel Method:** The novelty of NewtonNet is emphasized by all reviewers, particularly for its unique approach in incorporating graph homophily/heterophily and using Newton interpolation techniques in spectral GNNs. (All reviewers)

**Comprehensive Experiments:** The proposed NewtonNet demonstrates superior performance in extensive experiments, showcasing its efficacy across different datasets and domains. (Reviewers UTx2, zWto, and 7Vz6)

We have submitted an updated version of the manuscript in PDF, incorporating the suggestions provided by the reviewers. The following are our detailed responses to each weakness and question. We hope the weaknesses and questions are well addressed. Please let us know if you have further questions.

---

### Meta-Review · Area_Chair_wqpZ · 2023-12-15

**Metareview:**

The paper presents NewtonNet, a spectral Graph Neural Network framework that uses Newton interpolation for learning spectral filters, informed by graph homophily/heterophily levels, showing enhanced performance on diverse datasets. However, some concerns by the reviewers have not been fully addressed.

**Justification For Why Not Higher Score:**

There are some concerns by the reviewers about reproducibility.

**Justification For Why Not Lower Score:**

NA

---

### Decision · Program_Chairs · 2024-01-16

Reject